# A Hepatitis C virus genotype 1b post-transplant isolate with high replication efficiency in cell culture and its adaptation to infectious virus production in vitro and in vivo

**Christian Heuss[1]**, **Paul Rothhaar[1]**, **Rani Burm[2]**, **Ji-Young Lee[3]**, **Philipp Ralfs[1]**,
**Uta Haselmann[3]**, **Luisa J. Ströh[4]**, **Ombretta Colasanti[1]**, **Cong Si Tran[1]**, **Noemi Schäfer[1]**,
**Paul Schnitzler[5]**, **Uta Merle[6]**, **Ralf Bartenschlager[3,7,8]**, **Arvind H. Patel[9]**, **Frederik Graw[10,11]**,
**Thomas Krey[4,12,13,14,15]**, **Vibor Laketa[5,7]**, **Philip Meuleman[2]**, **Volker Lohmann[1,7]** *

1 Department of Infectious Diseases, Molecular Virology, Section virus-host interactions, Heidelberg
University, Heidelberg, Germany, 2 Laboratory of Liver Infectious Diseases, Ghent University, Gent,
Belgium, 3 Department of Infectious Diseases, Molecular Virology, Heidelberg University, Heidelberg,
Germany, 4 Institute of Virology, Hannover Medical School, Hannover, Germany, 5 Department of Infectious
Diseases Virology, University Hospital Heidelberg, Heidelberg, Germany, 6 Department of Internal Medicine
IV, University Hospital Heidelberg, Heidelberg, Germany, 7 German Center for Infection Research, partner
site Heidelberg, Heidelberg, Germany, 8 Division Virus-Associated Carcinogenesis, German Cancer
Research Center (DKFZ), Heidelberg, Germany, 9 MRC-University of Glasgow Centre for Virus Research,
Glasgow, United Kingdom, 10 BioQuant – Center for Quantitative Biology, Heidelberg University, Heidelberg,
Germany, 11 Interdisciplinary Center for Scientific Computing, Heidelberg University, Heidelberg, Germany,
12 Center of Structural and Cell Biology in Medicine, Institute of Biochemistry, University of Lübeck, Lübeck,
Germany, 13 Centre for Structural Systems Biology (CSSB), Hamburg, Germany, 14 German Center for
Infection Research (DZIF), Partner Site Hamburg-Lübeck-Borstel-Riems, Lübeck, Germany, 15 Cluster of
Excellence RESIST (EXC 2155), Hannover Medical School, Hannover, Germany

☯ These authors contributed equally to this work.
* Volker.Lohmann@med.uni-heidelberg.de

doi.org/10.1371/journal.ppat.1010472

UNITED STATES

**Data Availability Statement:** All data are available
in the main text or the supplementary materials.
Sequences of GLT1 and GLT1cc are available

## Abstract

Hepatitis C virus (HCV) is highly diverse and grouped into eight genotypes (gts). Infectious
cell culture models are limited to a few subtypes and isolates, hampering the development
of prophylactic vaccines. A consensus gt1b genome (termed GLT1) was generated from an
HCV infected liver-transplanted patient. GLT1 replicated to an outstanding efficiency in
Huh7 cells upon SEC14L2 expression, by use of replication enhancing mutations or with a
previously developed inhibitor-based regimen. RNA replication levels almost reached JFH-
1, but full-length genomes failed to produce detectable amounts of infectious virus. Long-
term passaging led to the adaptation of a genome carrying 21 mutations and concomitant
production of high levels of transmissible infectivity (GLT1cc). During the adaptation, GLT1
spread in the culture even in absence of detectable amounts of free virus, likely due to cell-
to-cell transmission, which appeared to substantially contribute to spreading of other iso-
lates as well. Mechanistically, genome replication and particle production efficiency were
enhanced by adaptation, while cell entry competence of HCV pseudoparticles was not
affected. Furthermore, GLT1cc retained the ability to replicate in human liver chimeric mice,
which was critically dependent on a mutation in domain 3 of nonstructural protein NS5A.
Over the course of infection, only one mutation in the surface glycoprotein E2 consistently

under GenBank accessions OM222702 and OM222703 respectively.

**Funding:** This work was funded by grants from the Deutsche Forschungsgemeinschaft (DFG) LO1556/ 4-2 (278191845), TRR179 (272983813) and TRR209 (314905040) as well as a grant from the German Center for Infection Research (DZIF) (TTU 05.821) to VoL; grants from the Deutsche Forschungsgemeinschaft (DFG) TRR179 (272983813) and TRR209 (314905040) as well as grants from the German Center for Infection Research (DZIF) TTU 05.821 and TTU 05.712 to RBa; a UK Medical Research Council grant (MC_UU12014/2) to AHP, a Deutsche Forschungsgemeinschaft (DFG) grant (SFB 900, within project B10 (158989968)) to TK, a grant from the Chica and Heinz-Schaller Foundation to FG and grants from the Ghent University Special Research Fund (UGent BOF) as well as an Excellence of Science grant from the Research Foundation – Flanders (FWO) and FNRS to PM. The funders had no role in study design, data collection and analysis, decision to publish, or preparation of the manuscript.

**Competing interests:** The authors have declared that no competing interests exist.

reverted to wildtype, facilitating assembly in cell culture but potentially affecting CD81 interaction in vivo.

Overall, GLT1cc is an efficient gt1b infectious cell culture model, paving the road to a rationale-based establishment of new infectious HCV isolates and represents an important novel tool for the development of prophylactic HCV vaccines.

## Author summary

Chronic HCV infections remain an important global health issue, despite the availability of highly efficient therapies. So far no protective vaccine is available, which is in part due to the high divergence of HCV variants and the limited possibility to mirror this genetic diversity in cell culture. It has been proven particularly difficult to grow infectious virus in cell culture, requiring extensive adaptation with multiple mutations, which in turn affect infectivity of the adapted variants in vivo. Here we have isolated a genotype 1b variant from a very high titer serum of a patient after liver transplantation (German Liver Transplant 1, GLT1), showing an outstanding genome replication efficiency in cultured hepatoma cells. We were able to adapt this isolate to production of infectious virus, therefore generating a novel full-replication cycle cell culture model for the highly prevalent HCV genotype 1b. Despite multiple mutations required, adapted GLT1 was still infectious in vivo. GLT1 therefore is an important development facilitating future efforts in vaccine development.

## Introduction

Worldwide more than 71 million people are chronically infected with the Hepatitis C virus (HCV) [1] resulting in a high risk to develop severe liver disease and hepatocellular carcinoma [2]. Despite the availability of highly efficient therapies based on direct-acting antivirals (DAA) [3], the World Health Organization classified HCV infections as a public health threat [4]. HCV belongs to the genus *Hepacivirus* in the family *Flaviviridae*. Due to its genetic heterogeneity, the virus is classified into eight genotypes that differ in their nucleotide sequence by up to 30%, representing a major hurdle for the development of prophylactic vaccines [5–8]. Genotype 1, and the subtypes 1a and 1b, are most prevalent worldwide [9]. HCV is an enveloped positive-strand RNA virus, comprising a genome of approximately 9,600 nucleotides that encodes a single polyprotein flanked by two non-translated regions. The polyprotein is processed by cellular and viral proteases into three structural proteins (core, E1 and E2) and seven non-structural proteins (p7, NS2, NS3, NS4A, NS4B, NS5A and NS5B). The structural proteins core (capsid) and E1/E2 (envelope glycoproteins) are physical components of the virus particle. p7 is a viroporin important for virus release and NS2 is part of the NS2-3 autoprotease and contains three N-terminal transmembrane segments vital for virion morphogenesis. NS3, NS4A, NS4B, NS5A and NS5B are essential and sufficient components of the viral replicase, all contributing to the generation of membranous replication organelles designated the membranous web and contributing to viral RNA synthesis, but also to virion assembly. NS3 harbors an N-terminal protease activity responsible for the cleavage of all NS3-5B junctions with its co-factor NS4A and a C-terminal helicase. NS4B is a key factor for the biogenesis of the membranous web. NS5A is a phosphoprotein proposed as a key regulator of RNA replication and assembly. NS5B is the viral RNA-dependent-RNA polymerase (RdRp) (reviewed in [10]).

Even more than 30 years after the discovery of HCV, it remains challenging to replicate patient-derived wildtype (WT) viral isolates in cultured cells. The first cell culture models based on subgenomic gt1b replicons (prototype isolate Con1) in the hepatocellular carcinoma (HCC) derived cell line Huh7 revealed the need for replication enhancing mutations to obtain robust RNA replication [11,12]. These mutations were also a prerequisite for the establishment of replicons of almost all other genotypes (1a, 3a, 4a, 5a, 6a) (reviewed in [13]). The only exception so far is the gt2a isolate JFH-1, which is capable of highly efficient RNA replication in Huh7 [14]. Meanwhile, the need for replication enhancing mutations has been overcome to some extent. First, reconstitution of SEC14L2 expression, a lipid transport protein expressed in human hepatocytes but not in Huh7, facilitated replication of WT isolates of all genotypes [15], albeit with varying efficiency [16]. Second, the identification of the mechanisms underlying replication enhancing mutations in NS5A (prototype: S2204I/R, referring to the amino acid position in the polyprotein of Con1) and NS5B (R2884G) [17], allowed the development of an inhibitor regimen termed PCi, based on the combined chemical inhibition of Phosphatidylinositol-4 kinase III alpha (PI4KA) and Casein Kinase Ia (CKIα), enabling RNA replication of gt1b isolates by compensating for the overexpression of PI4KA in HCC-derived cells, compared to human hepatocytes [18].

The second and even more demanding hurdle relates to the difficulty in generating infectious virus in cell culture. Since most initially identified replication enhancing mutations impaired or abrogated virus production [19], resulting in attenuation in vivo [20], it required the JFH-1 isolate to allow producing low amounts of infectious virus in cell culture [21]. However, here titer enhancing mutations were necessary for robust virus production [22] or chimeric genomes encoding the structural proteins, p7 and NS2 of another gt2a isolate (J6, [23]), either entirely (J6/JFH-1, [24]) or in parts (JC1, [25]). Later, a gt1a genome containing five replication and titer enhancing mutations was established, generating infectious virus with limited efficiency [26]. Meanwhile, additional genomes capable of infectious virus production based on gt1a, gt2a-c, gt3a, gt4a and gt6a have been generated ([27,28] and reviewed in [29]). Most of them were obtained using JFH-1-based chimeras as a starting point, building on repetitive passaging and recloning cycles, finally resulting in numerous replication- and titer enhancing mutations (reviewed in [29]). However, even though gt1b requires only one replication enhancing mutation and is effectively stimulated by SEC14L2 expression or PCi treatment [16,18], only recently two gt1b isolates could be adapted to efficient virus production in cell culture, again using JFH-1 based chimeras as a starting point [30,31].

This study aimed at establishing a novel gt1b isolate capable of efficient virus production in cell culture. Starting from a high titer serum of a post-transplant patient, we generated a consensus genome termed GLT1 with RNA replication capacity close to JFH-1. Infectious virus production required more than a year with a total of 118 cell passages and 27 supernatant transfers. The cloned GLT1cc genome contains 21 mutations, facilitating efficient virus production and spread, still retaining infectivity in vivo. GLT1cc therefore might greatly facilitate the development of a protective vaccine.

## Results

### Generation of GLT1 and analysis of RNA replication efficiency in cell culture

We recently established a drug regimen based on pharmacological inhibition of PI4KA and CKIα (designated PCi) enhancing RNA replication of gt1b WT isolates by approximately 100-fold, allowing infection of Huh7 hepatoma cells with several patient-derived gt1b sera [18]. One serum with an exceptionally high viral load (>100,000,000 IU/ml) gave rise to a

particularly strong increase in RNA replication upon PCi treatment (termed "Serum 1" [18]), indicating a viral isolate with exceptional replication efficiency. We therefore PCR-amplified the viral genomic sequences from infected Huh7 cells and generated a consensus sequence based on direct sequencing of the PCR products. In case of equivocal sequencing data, which affected the amino acid sequence only at two positions (aa849 in NS2 and 2760 in NS5B), we opted to incorporate the respective variant with higher frequency. This novel gt1b isolate was termed German Liver Transplant 1 (GLT1), since the serum was obtained from a patient after liver transplantation. Alignment of the GLT1 coding sequence with a general gt1b consensus sequence (Fig 1A, upper panel) or the widely used gt1b isolate Con1 [11] (Fig 1A, lower panel) revealed 95% and 94% identity, respectively. A phylogenetic tree of 358 available gt1b full-length genome sequences further demonstrated that the GLT1 isolate is located in one of the main branches of this subtype (Fig 1B).

We next aimed to characterize GLT1 in terms of its RNA replication efficiency as well as its ability to produce infectious virus in cell culture. To this end, we generated a synthetic consensus sequence for construction of a subgenomic reporter replicon and a full-length genome. Since we did not amplify the sequence of the genome termini due to their high degree of conservation, nt1-82 of the 5'UTR and the 3'terminal 31 nucleotides of the 3'UTR were taken from Con1. We first determined RNA replication efficiency in comparison to two isolates widely used in the field, representing experimental gold-standards so far (Fig 1C–1F and S1–S3 Figs): JFH-1 (gt2a), the only HCV isolate replicating efficiently in cell culture without any modification, and Con1 (gt1b) [11], a prototype isolate more closely representing the phenotype of all other cloned HCV isolates, requiring either PCi treatment [18], SEC14L2 expression (S1A Fig; [15]) or replication enhancing mutations for efficient RNA replication [12,17,32]. To assess RNA replication efficiency, we transfected luciferase reporter replicons (S1B Fig) of the three isolates into Huh7 cells (S2 Fig) and two highly permissive Huh7 subclones, Huh7-Lunet (Fig 1C–1F and S1C Fig, [18]) and Huh7.5 (S3 Fig; [33]), implementing all replication enhancing conditions accessible so far and using firefly luciferase activity as a quantitative measure. While GLT1 replication was only slightly increased in naïve Huh7-Lunet cells compared to Con1 (Fig 1C and 1D, black lines), stimulation by PCi treatment was far more pronounced (Fig 1C, green lines), almost reaching JFH-1 levels at 72 h after transfection. A similar increase was obtained by ectopic expression of SEC14L2 (Fig 1D, red lines; S1A Fig). Combination with PCi treatment had a slightly additive effect on Con1 and GLT1, but confirmed the overall 100-fold higher replication efficiency of GLT1 (Fig 1E). JFH-1 replication was not affected by SEC14L2 expression and slightly reduced upon PCi treatment, due to its different requirement for PI4P [18].

We further compared the impact of a set of well characterized replication enhancing mutations on Con1 and GLT1 (Fig 1F, [17]). Here, replication enhancement of GLT1 was the highest for a mutation in NS4B (mut4B, K1846T), following overall the same pattern as Con1, but again with 100-fold higher efficiency reaching a similar level as SEC14L2 combined with PCi (Fig 1E, blue lines). Combinations of adaptive mutations with SEC14L2 expression did not further increase replication levels of the mut4B variant (S1C Fig). Similar results were obtained in Huh7 (S2 Fig) and Huh7.5 cells (S3 Fig), albeit with overall slightly lower efficiency.

In conclusion, we identified and cloned a novel genotype 1b isolate (GLT1) from a liver transplant patient with about 100-fold higher replication efficiency than the closely related prototype isolate Con1. Efficient GLT1 WT replication in Huh7-derived cells required support by expression of SEC14L2, or by a previously established inhibitor cocktail, with some additive effects. A replication enhancing mutation in NS4B resulted in highest replication levels.

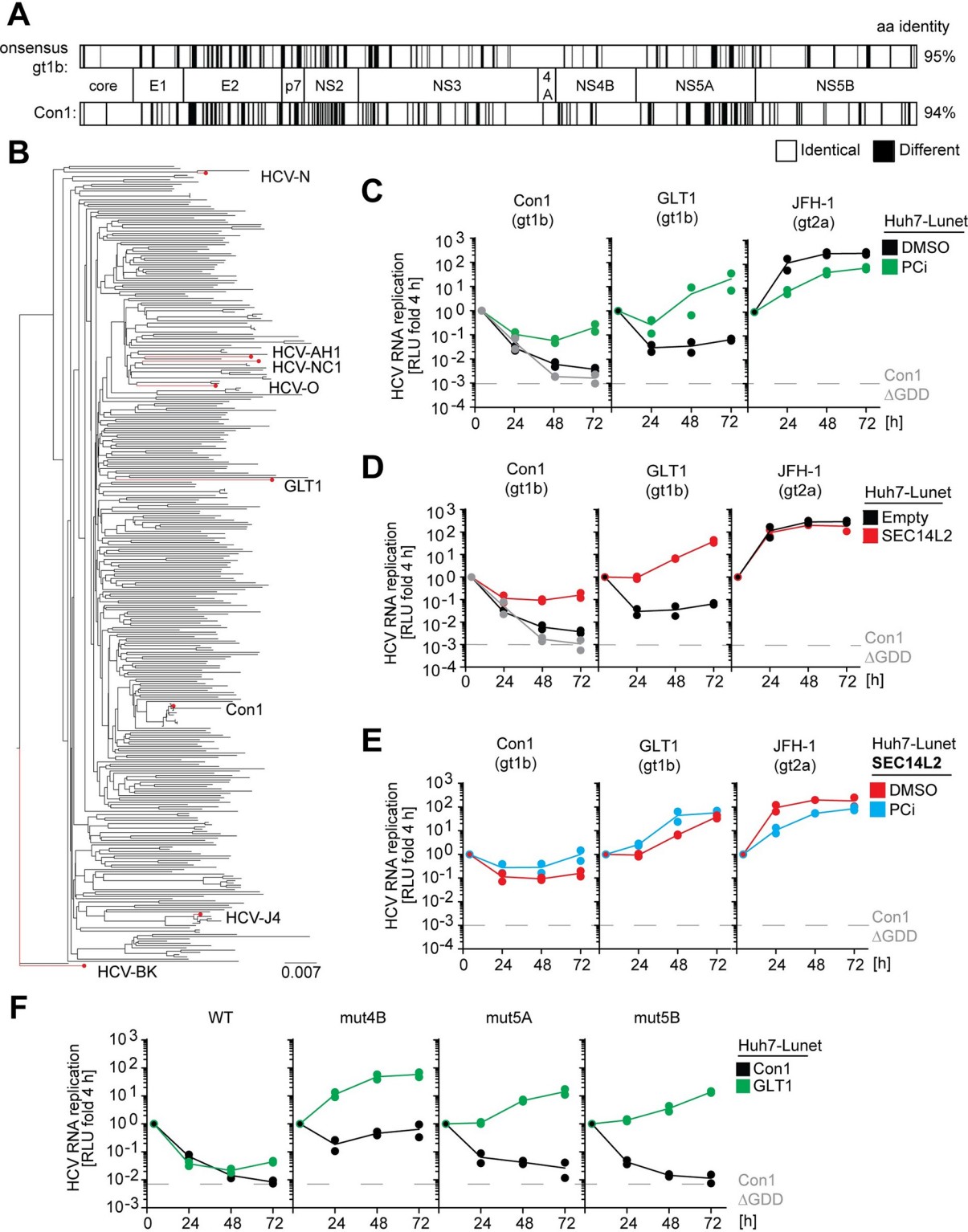

**Fig 1. Phylogeny and RNA replication efficiency of GLT1 in Huh7-Lunet cells compared to other HCV isolates.** (**A**) Alignment of the GLT1 amino acid sequence compared to a gt1b consensus (upper part) or Con1 (lower part). Black lines indicate differences. (**B**) Phylogenetic tree of gt1b isolates available in ViPR [80]. Some selected viral variants available and characterized as molecular clones are highlighted (GenBank accession numbers: HCV-N: AAB27127, HCV-AH1: BAG31965, HCV-NC1: BAM29294, HCV-O: BAD91386, Con1: CAB46677, HCV-J4: BAA01583, HCV-BK: AAA72945). (**C-F**) Replication efficiency of GLT1 compared to Con1 and JFH-1, using different replication enhancing

conditions. Huh7-Lunet cells were transfected with subgenomic reporter replicons of the indicated isolates or mutants. Luciferase activity in cell lysates (RLU) was quantified as a correlate of RNA replication efficiency at the given time points and normalized to 4 h. Cells were either stimulated by PCi treatment (C,E) and/or SEC14L2 expression (D,E) compared to DMSO treatment or empty vector transduction, respectively. (**F**) Replication enhancement of GLT1 (green lines) or Con1 (black lines) by mutations in NS4B (K1846T), NS5A (S2204R) or NS5B (R2884G). (C-F) A replication deficient Con1 variant (Con1ΔGDD) was used as a negative control for replication and the respective luciferase level at 72 h is indicated by a dashed grey line in all diagrams and shown in a time-course in the Con1 panels. The data are the mean values from two independent experiments shown as individual data points with two technical replicates each.

## Morphology of the replication organelles of GLT1

We next explored whether changes in the ultrastructure of replication organelles might explain the outstanding RNA replication competence of GLT1. This so-called membranous web mainly consists of double membrane vesicles (DMVs) with an average diameter of ca. 200 nm [34,35]. We first analyzed DMV abundance and size in cells transiently transfected with GLT1 replicons under all conditions providing sufficient replication efficiency for thorough quantification (PCi treatment, SEC14L2 expression or replication enhancing mutation mut4B; Fig 2A–2C). We used a reporter system based on the expression of GFP with a nuclear translocation signal fused to the membrane anchor of MAVS (MAVS-GFP-NLS, [18,36]) harboring the NS3-4A cleavage site. This allowed for the identification of HCV positive cells by nuclear GFP localization in correlative light and electron microscopy (CLEM). The DMV abundance was very variable due to the random choice of areas, but overall comparable (Fig 2A). The average DMV diameter ranged from 129–145 nm, which seemed relatively small compared to Con1 in a previous study (ca. 180 nm [18]). To exclude the impact of the replication enhancing conditions and changes in the protocol with potential impact on DMV diameter, we expressed GLT1 NS3-5B WT in naïve Huh7-Lunet T7 cells and found similar DMV diameters, which were moderately increased by SEC14L2 expression (157 nm and 178 nm, respectively, S4A and S4B Fig) and indeed significantly smaller than for Con1 (230 nm, S4A Fig). Since reduced DMV diameters were previously associated with reduced PI4P concentrations [18,35], we compared the level of PI4KA activation by expression of NS3-5B of JFH-1, Con1 and GLT1 (Fig 2D and 2E). Indeed, PI4P concentrations in HCV positive cells were moderately but significantly reduced in case of GLT1, pointing to a reduced level of PI4KA activation, similar but less pronounced compared to a class of replication enhancing mutations (mut5A and mut5B; [18]). The reduced level of PI4KA activation by GLT1 is therefore the likely cause of the observed changes in DMV phenotype and the potential reason of the increased replication of GLT1 compared to Con1 WT in naïve Huh7 cells (Fig 1C and 1D, black lines).

In sum, replication organelles of GLT1 WT contained smaller DMVs compared to other HCV isolates, in line with a reduced capacity to activate PI4KA. This might explain in part the higher replication capacity compared to related isolates.

## Full-length GLT1 requires extensive adaptation to generate transmissible infectivity

Next, we assessed if the GLT1 full-length genome was capable of producing infectious virus particles in Huh7 cells. We again used either PCi treatment or ectopic SEC14L2 expression to stimulate WT virus replication, including Con1 as a reference gt1b isolate. In addition, we used GLT1-mut4B since the K1846T mutation in Con1, in contrast to other major adaptive mutations, is known not to interfere with infectious virion production [19,20]. As a reference for efficient viral replication and virion release, we included the gt2 chimeric J6CF/JFH-1 genome, called JC1 [25]. In vitro transcribed full-length genomes were transfected into Huh7-Lunet cells and the intra- or extracellular amount of viral core protein was quantified as a

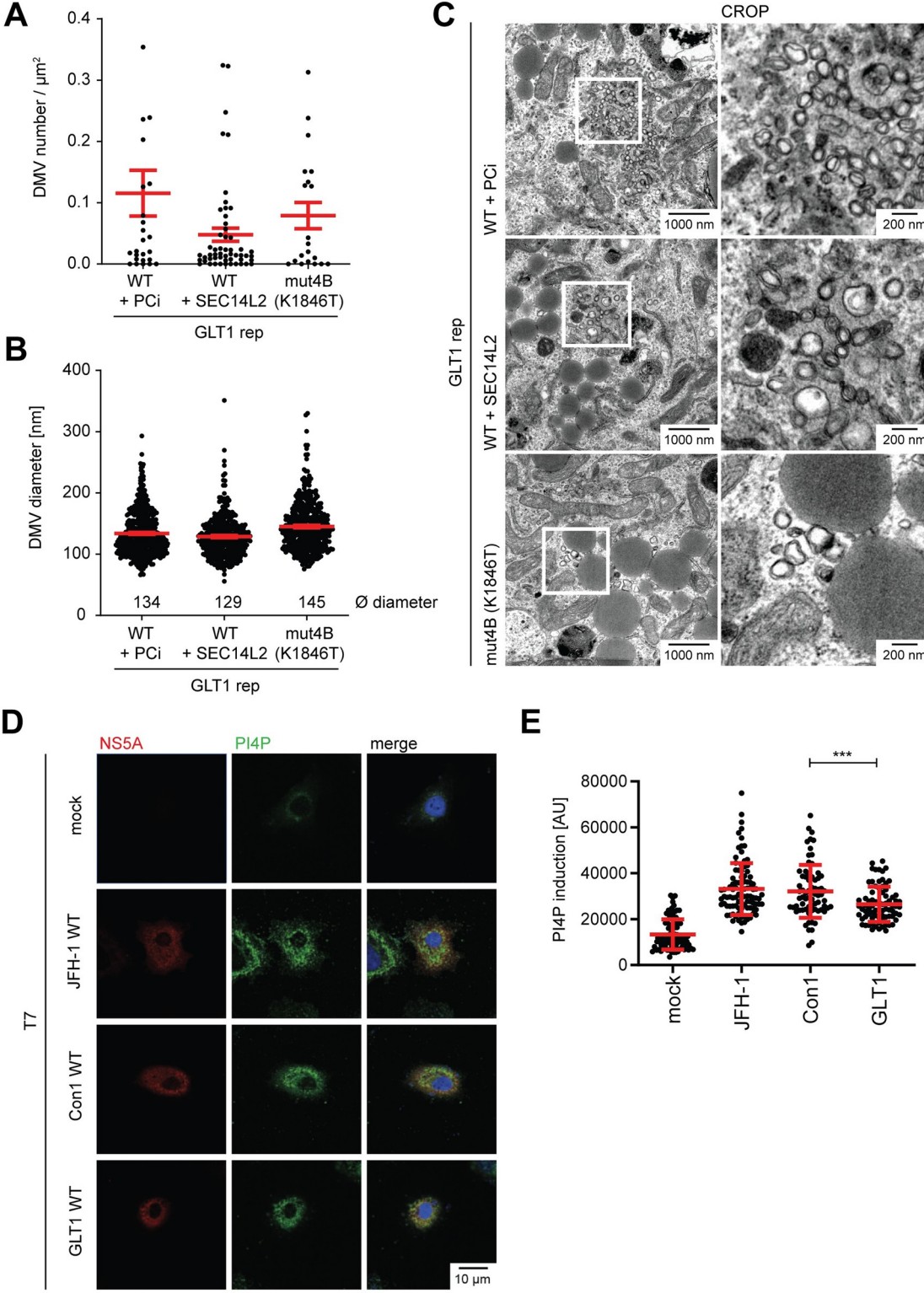

**Fig 2. Ultrastructural analysis of membrane rearrangements induced by the GLT1 replicon using correlative light and electron microscopy (CLEM).** Huh7-Lunet CD81 MAVS-GFP-NLS cells were transfected with GLT1 WT reporter replicons (GLT1rep) or a mutant encoding a replication enhancing mutation in NS4B (mut4B, K1846T). In case of GLT1rep WT, replication was enhanced either by SEC14L2 expression or PCi treatment. The cells were fixed 48 h post electroporation and the nuclear GFP signal was used to identify positive cells for further analysis. DMV profiles were analyzed using TEM images taken at x4k

magnification. For systematic random sampling, ~100 μm$^2$ rectangle areas were placed on a whole cell image and DMVs were counted. At least 6 positive cells were counted in each sample. (**A**) The number of DMVs per μm$^2$. (**B**) The diameter of DMVs. (**C**) Representative images of each condition. (**D,E**) Huh7-Lunet T7 cells were transfected with plasmids encoding NS3-5B of Con1 WT, GLT1 WT or JFH-1 WT and fixed 24 h post transfection. (D) Total NS5A and PI4P was detected by immunofluorescence analysis using monoclonal antibody 9E10 (red) and anti-PI4P (green), respectively. The nuclei were stained with DAPI. (E) Quantification of the PI4P signal intensity in 71–95 cells using Fiji. Each dot represents one cell. Results shown are mean values from two independent experiments. Unpaired two-tailed Student's t-test was used to determine statistical significance (*** = p ≤ 0.001).

correlate of RNA replication efficiency and virion production, respectively (Fig 3A). Differences in intracellular core levels for the different variants and conditions were consistent with the data of luciferase reporter replicons (Fig 1C–1F). Extracellular core levels were by far the highest in case of JC1 and almost undetectable for Con1, as expected. In case of GLT1 WT, high intracellular core protein levels correlated with increase in the extracellular core protein for PCi treatment and SEC14L2 expression, as well as for mut4B, but remained 100-fold lower than JC1. A similar increase was observed in Huh7.5 cells, albeit with slightly lower efficiency (S5B Fig). The level of extracellular core protein did not increase by combined PCi treatment and SEC14L2 expression (Fig 3A), nor by the generation of a chimeric gt1b genome containing the structural proteins and parts of NS2 of Con1 and the replication module of GLT1 (Con1/C3GLT1, S5C Fig). To assess the presence of infectious virus, we used Huh7-Lunet CD81 cells stably expressing the MAVS-GFP-NLS reporter [18] allowing for a sensitive detection of infection events in living cells. For transfer of supernatants of Con1 and GLT1 WT, we further expressed SEC14L2 in the recipient cell line. However, no infection events were detectable from supernatants at 72 h after transfection except for JC1 (Fig 3A and 3B and S5A Fig).

Strong increase of infectious virus levels for JFH-1 and other isolates was achieved by passaging of transfected cells, followed by supernatant passaging, resulting in the accumulation of titer enhancing mutations [29]. We tried this approach also for GLT1, using the most promising conditions (SEC14L2 expression and GLT1-mut4B) and including Con1 as a control (Fig 3C). Huh7-Lunet cells with or without SEC14L2 expression were transfected with the respective HCV variants and passaged twice a week. The status of each culture was monitored using the extracellular core protein level (Fig 3C) as well as the GFP localization. The level of extracellular core protein constantly declined in case of Con1 and GLT1 WT in SEC14L2 expressing cells and finally fell below the limit of detection at p5 and p15, respectively. Con1/C3GLT1 performed slightly better, still amounts of secreted core protein continuously declined and the passaging was stopped after p21 since no detectable amounts of infectious virus were found after transfer of supernatant (S5D Fig). GLT1-mut4B led to consistent secretion of core protein and transfer of concentrated supernatant of passage number 29 finally resulted in some small clusters of positive cells representing infection events (Fig 3D). The initially low number of positive cells increased upon passaging and we continued to transfer pure or concentrated supernatants after 3–5 cell passages to naïve reporter cells (Fig 3E). After 8 supernatant transfers and a total of 64 cell passages (p64.8, Fig 3E), the first infection events upon transfer of pure supernatant were obtained (Fig 3E). Interestingly, the initial number of infection events after each supernatant transfer did not change dramatically, but the velocity of spread within the culture, estimated by the number of GFP positive nuclei, increased rapidly within a few passages up to approximately 80% positive cells (Fig 3E), consistent with the dynamics of core-secretion (red lines). Since continuous passaging did not substantially increase the initial number of infected cells after supernatant transfer, nor the number of passages required to reach 80% positive cells further decreased, we stopped the experiment after 27 supernatant passages, including a total number of 118 cell passages (p118.27).

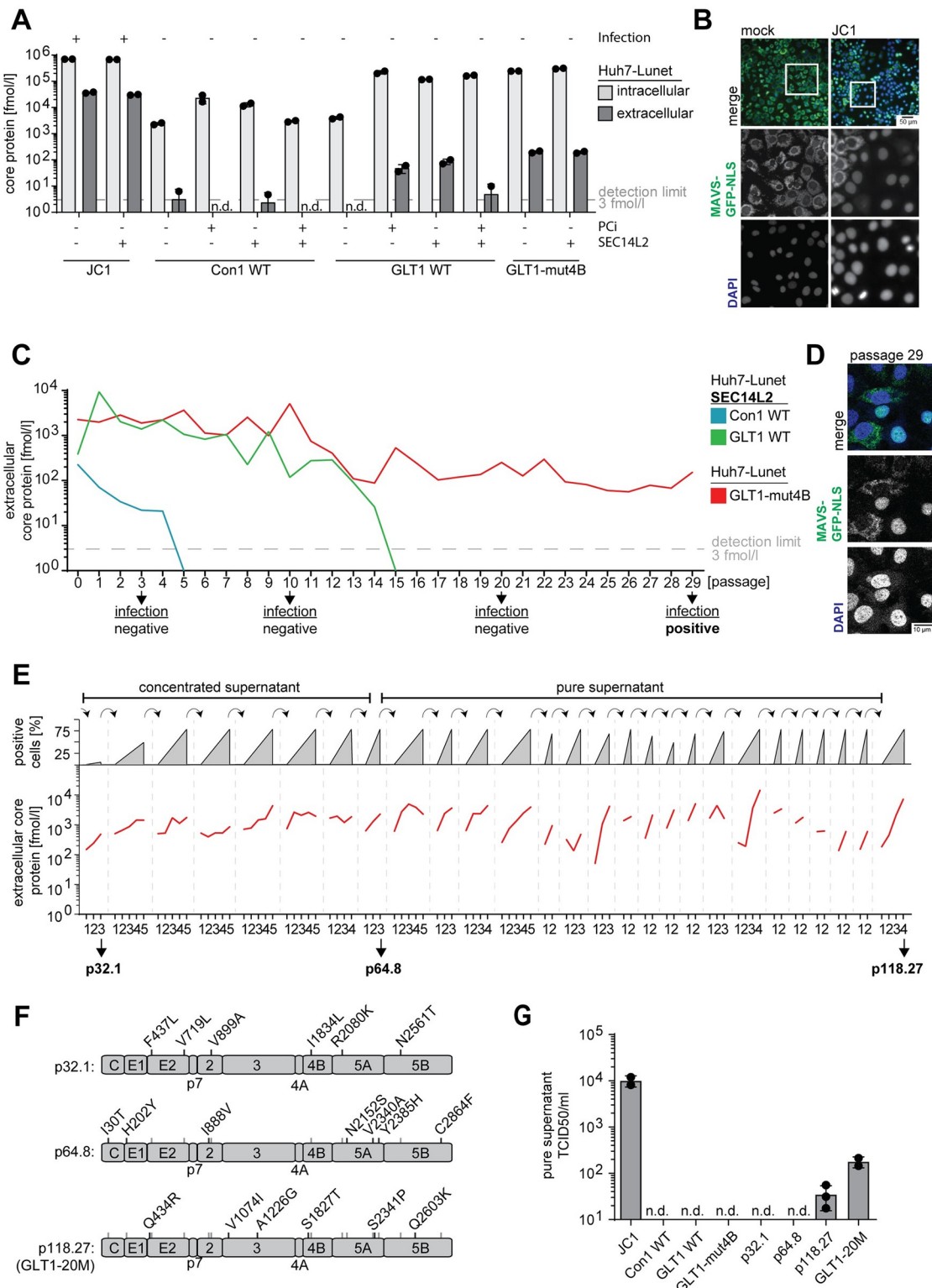

**Fig 3. Evaluation of virus production and passaging of GLT1.** (**A**) Detection of intra- and extracellular core protein after transfection of full-length virus genomes. Huh7-Lunet with or without SEC14L2 expression and/or with PCi treatment were transfected with the indicated HCV full-length genomes. 72 h post transfection, the concentration of core protein in lysate (intracellular, light grey) and supernatant (extracellular, dark grey) of one well of a 6-well dish was quantified by ELISA, as correlates of RNA replication and virus production, respectively. Shown are data from two independent experiments. The dashed grey line

indicates the detection limit of 3 fmol/l core protein; n.d. = not detectable. Successful infection upon transfer of supernatants to naïve cells is shown on top of each condition. (**B**) Example of infection events upon supernatant transfer to MAVS-GFP-NLS cells. Infection is identified by nuclear GFP signal. (**C**) Huh7-Lunet CD81 cells with or without SEC14L2 expression were transfected with Con1 (blue), GLT1 WT (green) or GLT1-mut4B (red) RNA and passaged. Before each passage, the amount of extracellular viral core protein was determined by ELISA. The dashed grey line represents the detection limit of 3 fmol/l core protein. Arrows indicate time points of infections experiments with concentrated supernatant using Huh7-Lunet CD81 MAVS-GFP-NLS with or without SEC14L2 expression. (**D**) Infection events after transfer of concentrated supernatant following p29. (**E**) Extracellular amount of viral core protein over the course of additional rounds of supernatant transfer and cell passaging for GLT1-mut4B in Huh7-Lunet CD81 MAVS-GFP-NLS cells. Each transfer event is indicated by the dashed grey lines, numbers below refer to the following cell passages. Three time points were highlighted with their total number of cell passages followed by the total number of supernatant passages (e.g. p118.27). Infections were either done with concentrated or pure supernatant as indicated on top. Grey triangles represent the approximate number of cells with a nuclear MAVS-GFP signal, shown in percent. (**F**) Consensus sequences derived by direct sequencing of RT-PCR products after p32.1 (first successful concentrated supernatant transfer), p64.8 (first pure supernatant transfer) and p118.27 (endpoint). Note that none of the changes reverted back to WT during passaging and were combined in a new synthetic genome termed GLT1-20M. (**G**) Determination of viral titers (TCID50/ml) from pure supernatant after either transfection of the indicated HCV full-length RNA or after the indicated passages of GLT1-mut4B. Shown are mean values and standard deviation of at least two independent experiments.

To identify potential titer enhancing mutations, we determined the consensus sequence of the viral quasispecies by direct sequencing of PCR products after the first passage of concentrated supernatant (p32.1), the first successful passage of pure supernatant (p64.8) and at the endpoint of the experiment (p118.27) (Fig 3F). The number of conserved mutations successively increased to 19 at p118.27, whereas K1846T was maintained, resulting in a total of 20 deviations compared to GLT1 WT. Mutations were spread over the entire polyprotein with one mutation in the core protein (I30T), four in the glycoproteins E1 and E2 (H202Y / Q434R / F437L / V719L), two in NS2 (I888V / V899A), two in NS3 (V1074I / A1226G), two in NS4B (S1827T / I1834L), five in NS5A (R2080K / N2152S / V2340A / S2341P / Y2385H) and three in NS5B (N2561T / Q2603K / C2864F). Due to this large number and the incremental increases in efficiency over the course of passaging, we opted to not study the titer enhancing effects of individual mutations. Instead, we generated a consensus clone of the dominant viral sequence found at p118.27, which we termed GLT1-20M. Transfection of in vitro transcribed homogenous GLT1-20M RNA resulted in $1.5 \times 10^2$ TCID50 per ml compared to $3 \times 10^1$ detected in the heterogenous population of supernatant p118.27 and $1 \times 10^4$ for JC1 (Fig 3G).

In conclusion, full-length GLT1, neither wildtype nor the mut4B genome, produced efficient levels of infectious virus in cell culture. However, after continuous rounds of cell and supernatant passages the viral genome acquired 19 potential titer enhancing mutations, generating a low but detectable level of transmissible virus.

## Characterization of GLT1 transmission mechanisms

Since GLT1 showed a rapid increase in the number of HCV positive cells (up to 80%) after a few passages, despite the low amounts of free virus in the supernatant, we aimed to further characterize the modes of transmission. In a continuous culture, new positive cells can occur by division of infected cells, by transmission of free virus or by cell-to-cell spread [37,38]. To allow quantification of transmission events, we established a co-culturing assay based on co-seeding of HCV positive cultures of MAVS-GFP-NLS cells with naïve cells harboring MAVS-mCherry-NLS in a ratio of 1:5. Therefore, cells with nuclear GFP signal could be classified as "donor", whereas cells with nuclear mCherry signal could be interpreted as "recipient", based on automated image analysis of whole wells at 24, 48 and 72 h after seeding (Fig 4A). Donor cultures were either chosen from the critical points of the passaging experiment (p32.1, p64.8 and p118.27, Fig 3E), or after transfection of in vitro transcripts of defined variants (GLT1-mut4B, GLT1-20M, JC1). Interestingly, the capacity of GLT1 to spread in cell culture

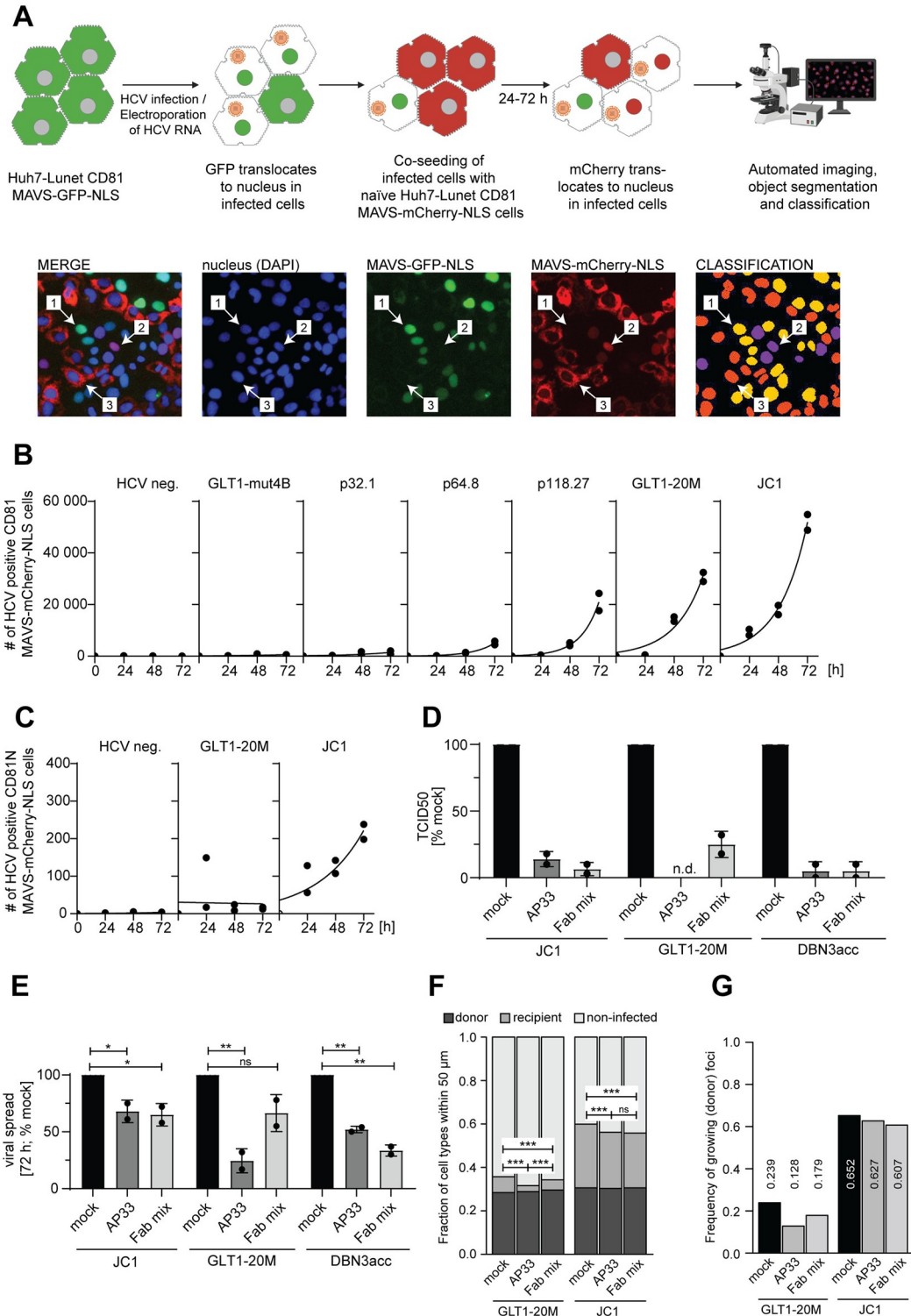

**Fig 4. Characterization of cell-free and cell-to-cell spreading properties.** (**A**) Schematic representation of the experimental procedure to analyze viral spread. Below are representative pictures of the individual fluorescent microscopy channels, a merge file of all fluorescent channels as well as an image representing the classification after the analysis using different colors (far right). Nuclei shown with a yellow color (see #1) represent cells with a nuclear GFP signal classified as "donor" cells (nuclear GFP). Purple nuclei represent "recipient" cells after virus transmission in co-culture (nuclear mCherry). Orange nuclei represent "non-infected" cells (cytoplasmic mCherry/GFP, nuclear DAPI). (**B,C**) Huh7-Lunet

CD81 MAVS-GFP-NLS cells transfected with full length HCV GLT1-mut4B, -20M or JC1 or obtained upon passaging of GLT1-mut4B (Fig 3E) were co-seeded with Huh7-Lunet MAVS-mCherry-NLS either (B) expressing CD81 or (C) a Huh7-Lunet variant sorted for low CD81 expression termed CD81N [38] in a ratio of 1:5 and incubated for 24–72 h. At the indicated time points, cells were fixed and the number of cells with a nuclear mCherry signal was determined via fluorescence microscopy followed by machine learning object classification. Black dots represent measurements for two independent experiments with the black line indicating the approximated mean time course. (**D**) Determination of the viral titer (TCID50/ml) after 72 h from concentrated supernatant of different HCV isolates. The viral stocks were incubated with 10 μg/ml of anti-E2 broadly neutralizing antibodies, using either AP33 or Fab mix or mock treated for 1 h at 37˚C prior to the infection of Huh7-Lunet CD81 MAVS-GFP-NLS. Shown are the mean and standard deviation of two independent experiments (black dots). (**E**) HCV positive Huh7-Lunet CD81 MAVS-GFP-NLS were co-seeded with Huh7-Lunet CD81 MAVS-mCherry-NLS and analyzed after 72 h as in (B,C) but in presence of 10 μg/ml of either AP33 or Fab mix. Shown are the mean and standard deviation of two independent experiments (black dots). Unpaired two-tailed Student's t-test was used to determine statistical significance (ns = not significant, * = p ≤ 0.05, ** = p ≤ 0.01). (**F**) Fraction of differently classified cell types within an area of 50 μm around a donor cell of GLT1-20M or JC1 treated with 10 μg/ml of anti-E2 broadly neutralizing antibodies, either AP33 or Fab mix, 72 hours after seeding. To assess significance, pairwise t-test on frequency of donor cells around spreader cells with p-values corrected for multiple comparisons was used (*** = p ≤ 0.001). (**G**) Frequency of individual donor (nuclear GFP) foci associated with at least one recipient (nuclear mCherry) cell within an area of 50 μm. This number shall represent an approximation of the likelihood of an individual donor cell infecting a non-infected cell in its surrounding. To check for significance, $\chi^2$-test of dependency between treatment and frequency of growing foci was performed (p<0.001 (GLT1-20M), p = 0.06 (JC1)).

increased with continuous passaging, similarly to the appearance of free transmissible virus (Fig 4B). The spreading efficiency and kinetics of the consensus genome GLT1-20M was further increased compared to the heterogenous HCV quasispecies present in p118.27 and remained only twofold reduced compared to JC1 (Fig 4B), despite the far lower titers (Fig 3G). The ability of viral spread for both genomes was widely dependent on the presence of CD81 as the co-culture with Huh7-Lunet CD81N cells, lacking CD81 expression [38,39], resulted in the absence of infection events in case of GLT1-20M (Fig 4C). In case of JC1, very low, but still detectable transmission rates were observed, in line with a previous report showing CD81 independent cell-to-cell spread using the same cell line [38] (Fig 4C). To differentiate between cell-free transmission and cell-to-cell spread, we next aimed to block cell-free transmission by the addition of neutralizing antibodies, comparing GLT1-20M with JC1 and the cell culture adapted gt3a isolate DBN3acc [40], generating similar titers as GLT1-20M in the absence of neutralizing antibodies (S6A Fig). We chose two different broadly neutralizing anti-E2 antibodies efficiently blocking infection: AP33 [41,42] and an equimolar mixture of Fab fragments derived from the two potent neutralizing antibodies AR3C [43] and HC84.1 [44] termed Fab mix. Indeed, both antibodies blocked infection of JC1 and DBN3acc with similar efficiency, preventing 80–90% of infection events (Fig 4D). For GLT1-20M, infection was undetectable upon AP33 treatment, whereas, Fab mix was far less efficient under these conditions (Fig 4D). In the co-culture setting, the antibodies were applied upon co-seeding of the cells and infection was evaluated 72 h later. Interestingly, the presence of neutralizing antibodies suppressed less than 50% of infection events in case of JC1, 30–40% in case of DBN3acc and inhibition ranged from 40% (Fab mix) to 80% (AP33) for GLT1-20M, in line with the varying neutralization capacity (Fig 4E). Overall, these data suggested that most of the GLT1-20M transmission events were dependent on E2 and could be blocked by anti-E2 neutralizing antibodies, arguing against a higher cell-to-cell spread efficiency compared to that of JC1 or DBN3acc.

Analysis of the spatial distribution at single cell resolution of donor and recipient cells confirmed the varying impact of the neutralizing antibodies for JC1 and GLT1-20M on the efficiency of viral spread. Despite similar cell densities, donor cells for GLT1-20M had on average a considerably lower frequency of recipient cells in their direct surrounding than donor cells for JC1 (Fig 4F). Furthermore, only 24% of all donor cells had at least one recipient cell within a radius of 50 μm indicating successful viral spread to neighboring cells, in contrast to 65% for

JC1 (Fig 4G). Supplementation of antibodies, especially AP33, reduced the frequency of these growing donor foci for GLT1-20M by nearly 50% (p<0.001, $\chi^2$-test of dependency between treatment and frequency of growing foci), while there was no effect for JC1 (p = 0.06, Fig 4G). As a result of the strong impact of the antibodies on GLT1-20M spread, the median distance between a donor cell and the next recipient cell was strongly increased (S6B Fig) in contrast to JC1, which generally experienced smaller distances between donor and recipient cells independent of the treatment conditions (S6C Fig). Thus, also the local analysis showed that GLT1-20M had a lower transmission capacity and was more effectively blocked by neutralizing antibodies than JC1. However, to clearly refer these observations to differences in specific transmission mechanisms (i.e., cell-free vs. cell-to-cell spread), a limitation of this analysis was the undefined time-lapse between transmission event and nuclear transfer of mCherry-NLS (16 h-48 h, S7A, S7B and S7E Fig and S1–S3 Movies), as well as the motility of the donor and recipient cells (S7C Fig and S4 Movie) and other unexpected events like the death of donor cells (S7D Fig and S5 Movie). Therefore, a cell-to-cell contact at the time of infection could not be formally excluded in some of the analyzed infection events.

In summary, acquired adaptive mutations for GLT1-20M mediated the ability to produce infectious virus and to spread in cell culture. Detailed analysis of the viral accessibility towards broadly neutralizing antibodies during viral spread did not support a higher cell-to-cell spreading efficiency of GLT1-20M compared to JC1 or DBN3acc.

## Generation of GLT1cc

In search for further ways to improve the efficiency of GLT1-20M in terms of viral titers, we analyzed the quasispecies of p118.27 to identify minor variants with promising potential, by sequencing a panel of individual cloned PCR products (S8A Fig). Surprisingly, three of five subclones contained the mutation N2415S, albeit direct sequencing of the PCR product, initially used to generate the GLT1-20M sequence, revealed only 10% of signal intensity compared to wildtype (S8B Fig). This site is located at the P5 position of the NS5A-NS5B cleavage site and a previous study found a titer enhancing mutation at P3, which was shown to slow down processing of NS5A-NS5B [22]. In addition, mutations at this [30] or at neighboring positions were also found in almost all cell culture adapted variants of other isolates [29]. We therefore reasoned that N2415S might have a titer enhancing effect in case of GLT1 as well. Interestingly, N2415S already increased the amount of secreted infectivity to low but detectable levels in case of GLT1-mut4B and enhanced titers of GLT1-20M by about 10-fold in every single experiment (Fig 5A). We thereby reached TCID50 values for GLT1-20M+N2415S around 10-fold below JC1 in Huh7-Lunet cells. For Huh7.5 cells, titers were substantially lower (S8C Fig), likely due to the reduced replication efficiency of gt1b in this subclone (Fig 1 compared to S3 Fig). The levels of secreted core protein and HCV genomic RNA were not substantially increased by addition of N2415S (Fig 5B and 5C), in contrast to the number of infected cells in a bulk culture infection (Fig 5D). At this point, we decided to stop further attempts to increase GLT1 titers and therefore termed GLT1-20M+N2415S as GLT1cc, in accordance with previous studies [24,29,45].

In summary, combining a replication enhancing mutation in NS4B with 19 conserved changes acquired during passaging and one intentionally added mutation at the C-terminus of NS5A dramatically enhanced the efficiency of infectious virus production by the GLT1 isolate. The resulting variant was designated GLT1cc.

## Mechanisms of titer enhancement

The changes acquired upon passaging were widespread across all viral proteins except p7 and NS4A (Fig 6A). Some variants were found in several published isolates, others were unique.

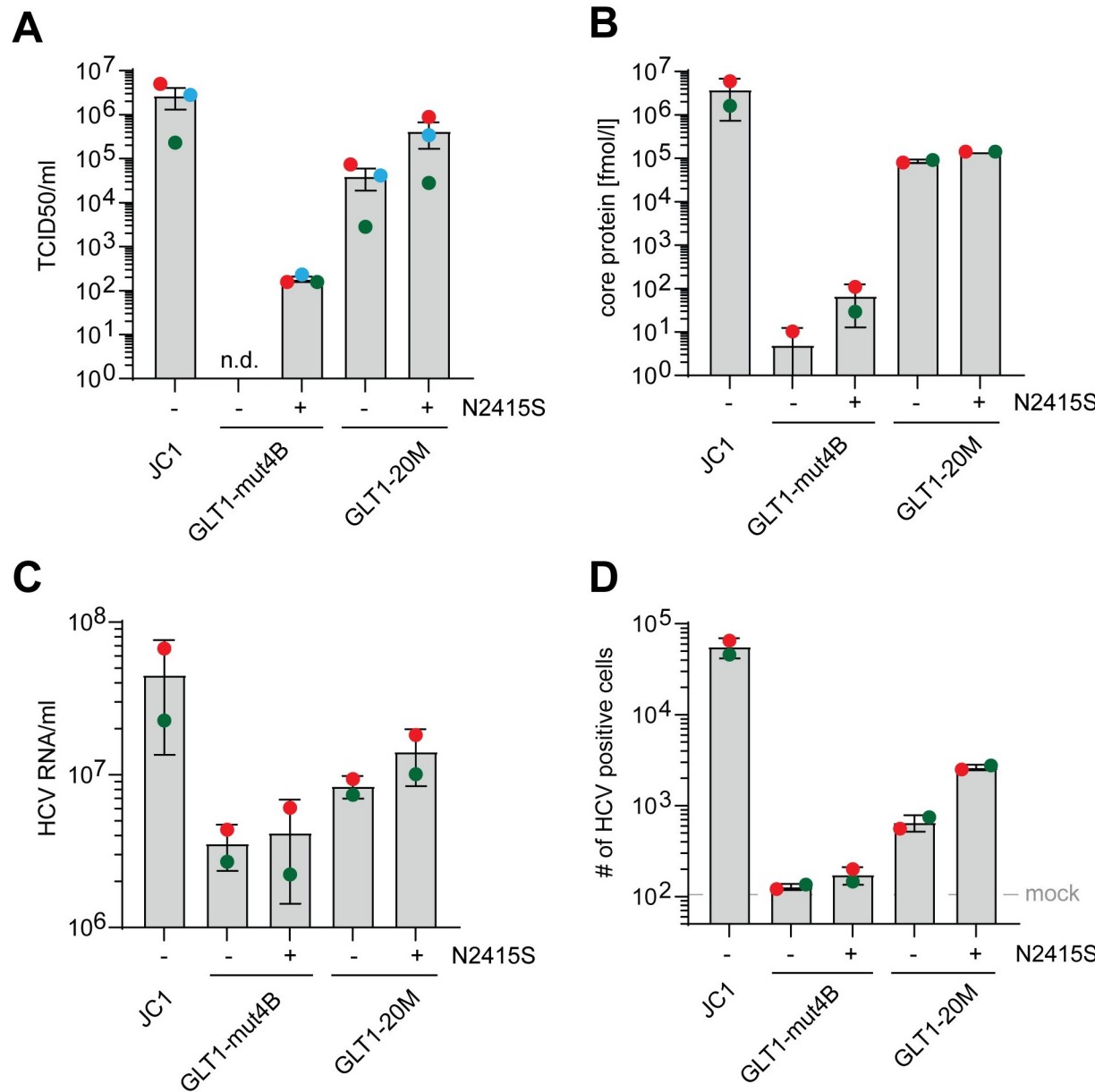

**Fig 5. Impact of N2415S on virion production of GLT1.** Huh7-Lunet CD81 MAVS-GFP-NLS were transfected with the indicated HCV genome variants, supernatants harvested at 24 and 48 h after transfection and concentrated 100-fold to determine (**A**) TCID50/ml on Huh7-Lunet CD81 MAVS-GFP-NLS cells, (**B**) core protein secretion by ELISA, (**C**) viral RNA using RT-qPCR and (**D**) the number of infected Huh7-Lunet CD81 MAVS-GFP-NLS 72 h after infection with 100 μl virus stock by classification with 2 color iLastik (infected / non-infected) based on GFP localization. Colors of data points represent one independent biological repetition across each sub-panel.

We therefore addressed the basic mechanisms underlying the improved efficiency of infectious virus production of GLT1cc by comparing RNA replication efficiency in a subgenomic replicon, entry efficiency using retroviral pseudoparticles decorated with the HCV glycoproteins (HCVpp) [46] and virus production in a JFH-1-based chimeric construct [25,47] (Fig 6B–6D). Since most mutations were found in the replicase proteins, we indeed found a clear increase in RNA replication kinetics for GLT1-20M and GLT1cc compared to GLT1-mut4B reporter

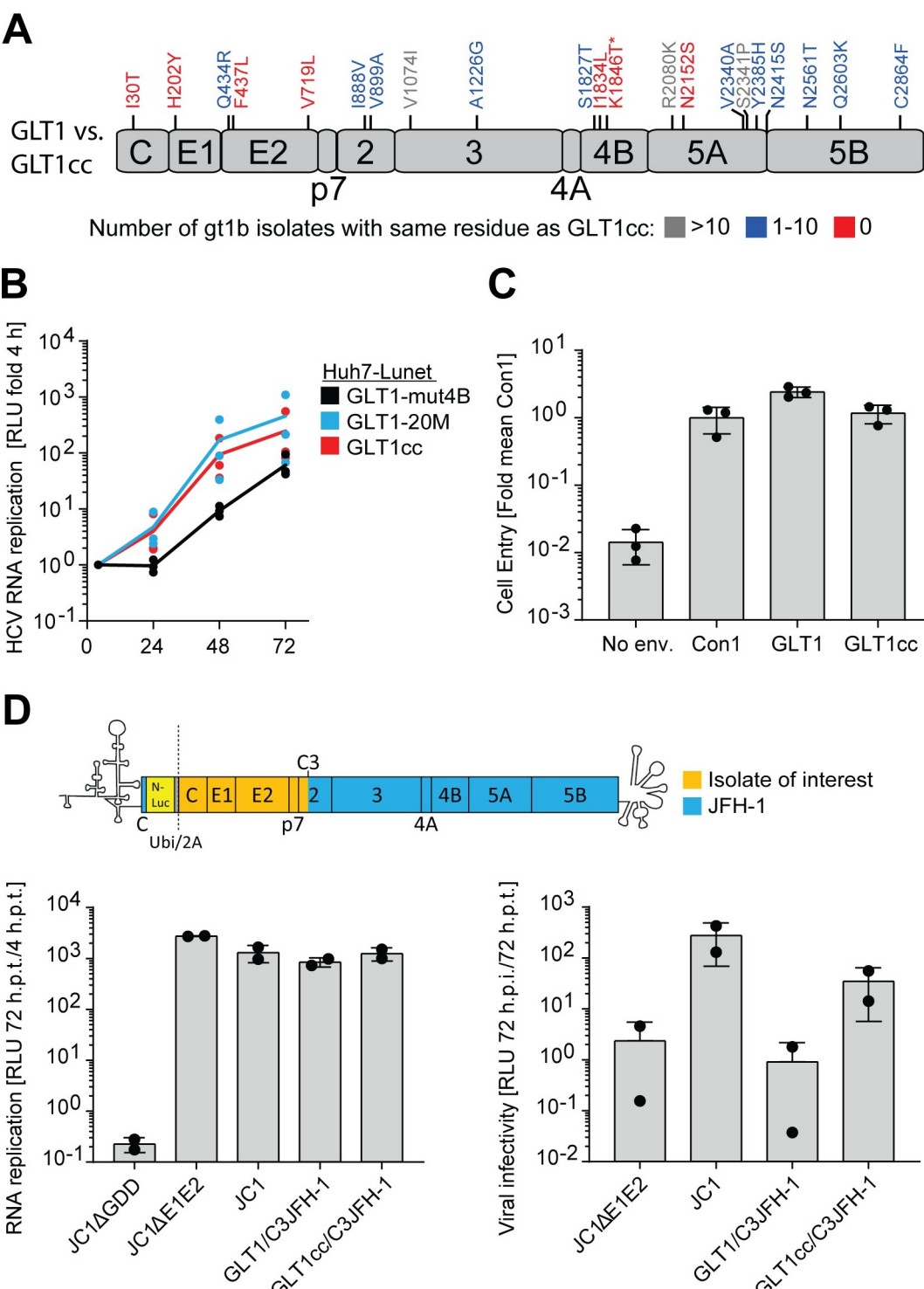

**Fig 6. Mechanisms of GLT1cc adaptation.** (**A**) Differences in amino acid sequence between GLT1 and GLT1cc and their abundance in other gt1b isolates are highlighted. Analysis based on 358 full-length HCV gt1b polyprotein sequences derived from ViPR. Asterisk indicates adaptive mutation in NS4B of GLT1-mut4B. (**B**) Huh7-Lunet cells were transfected with subgenomic reporter replicons of the indicated isolates. Luciferase activity in cell lysates (RLU) was quantified as a correlate of RNA replication efficiency at the given time points and normalized to 4 h. The data are the mean values from three independent experiments shown as individual data points with two technical replicates each. Unpaired two-tailed Student's t-test was used to determine statistical significance, no statistically significant difference between GLT1-mut4B and GLT1-20M

or GLT1cc respectively was found. (**C**) HCVpp bearing HCV envelope proteins of the indicated isolates on their surface were added to Huh7-Lunet CD81 cells. After 72 h, cells were lysed, and infectivity was determined based on a luciferase reporter in the HCVpp. Measurements were first normalized to the input determined via SG-PERT and then normalized to the resulting mean value of three replicates of the well-established Con1 isolate. Particles without envelope proteins (No env.) served as negative control. The data are the mean values from three independent experiments shown as individual data points with two technical replicates each. (**D**) Huh7-Lunet CD81 cells were transfected with the chimeric full-length HCV reporter construct depicted in the upper panel. It combines the sequence of the isolate of interest up to the C3 junction site in NS2 with the remaining part of the JFH-1 isolate [25]. Therefore, the two mutations in NS2 contained in GLT1cc were not included in the chimeric genomes. RNA replication was determined at 72 hours post transfection (h.p.t.) and the supernatant was used to infect naïve Huh7-Lunet CD81 cells. At 72 hours post infection (h.p.i.), cells were lysed, and infectivity was determined via luciferase measurement. A replication deficient JC1 variant (JC1ΔGDD) and a JC1 variant lacking envelope proteins (JC1ΔE1E2) were used as negative controls for replication and infection respectively. The data are the mean values from two independent experiments shown as individual data points with two technical replicates each.

replicons, suggesting that titer enhancement in part might be due to higher RNA replication efficiency (Fig 6B), although the differences did not reach statistical significance due to the high standard deviation between replicates. Still, this result did not exclude that changes in the NS proteins might further contribute to virion production by their reported functions in viral assembly (reviewed in [48]). Interestingly, when we replaced the structural protein coding region in the chimeric reporter virus genome of JCN2A [47] by the GLT1 counterparts, RNA replication was not affected, but production of infectious virions was reduced to the level of the negative control ΔE1E2 (Fig 6D). This was not unexpected, since such intergenotypic chimeras often suffer from incompatibility of replication and assembly modules [25,49], requiring further adaptation. However, the mutations found in GLT1cc partially rescued assembly of the chimeric genome, suggesting that these changes indeed facilitate the morphogenesis of infectious virions (Fig 6D). In contrast, entry efficiency of HCVpp was comparable between GLT1 and GLT1cc and similar to Con1 in Huh7-Lunet CD81 cells (Fig 6C). Due to the expected incremental contribution of individual mutations and the complexity of possible combinations, we did not further try to evaluate whether or not individual changes contributed to the phenotype.

## GLT1 replication in human liver chimeric mice

Finally, we evaluated the infectivity of GLT1 WT, GLT1-mut4B, GLT1-20M and GLT1cc in vivo, using homozygous uPA$^{+/+}$-SCID mice transplanted with primary human hepatocytes, rendering them permissive for HCV infection [50]. In case of GLT1 WT, we used the high-titer post-transplant patient serum as inoculum, due to its homogenous consensus sequence and the lack of efficient virus production in cell culture. Indeed, three out of three mice got infected and showed a high titer viremia for several weeks, with slightly different courses, which were within the regular variations found in this model (Fig 7A). We confirmed the HCV consensus sequence in total liver RNA of one of the mice sacrificed 8 weeks post infection by direct sequencing of RT-PCR products covering the whole HCV coding sequences to be identical with GLT1. Only at two positions (K1052R in NS3 and R1649K in NS4A) previous minor variants present in the inoculum became dominant (Fig 7B and S9 Fig). This result confirmed that GLT1 WT was infectious in vivo, supporting the entire viral replication cycle, albeit not generating detectable infectivity in cell culture.

Next, we aimed for a comparison of GLT1-mut4B, GLT1-20M and GLT1cc. Since infectious virus was not detectable in cell culture for GLT1-mut4B, we used the same amounts of 100-fold concentrated supernatants for the intrasplenical infection of three mice each (Fig 7C). For GLT1-mut4B, one mouse got infected and reached a continuously high titer, whereas one mouse died prior to the first blood withdrawal and one remained uninfected

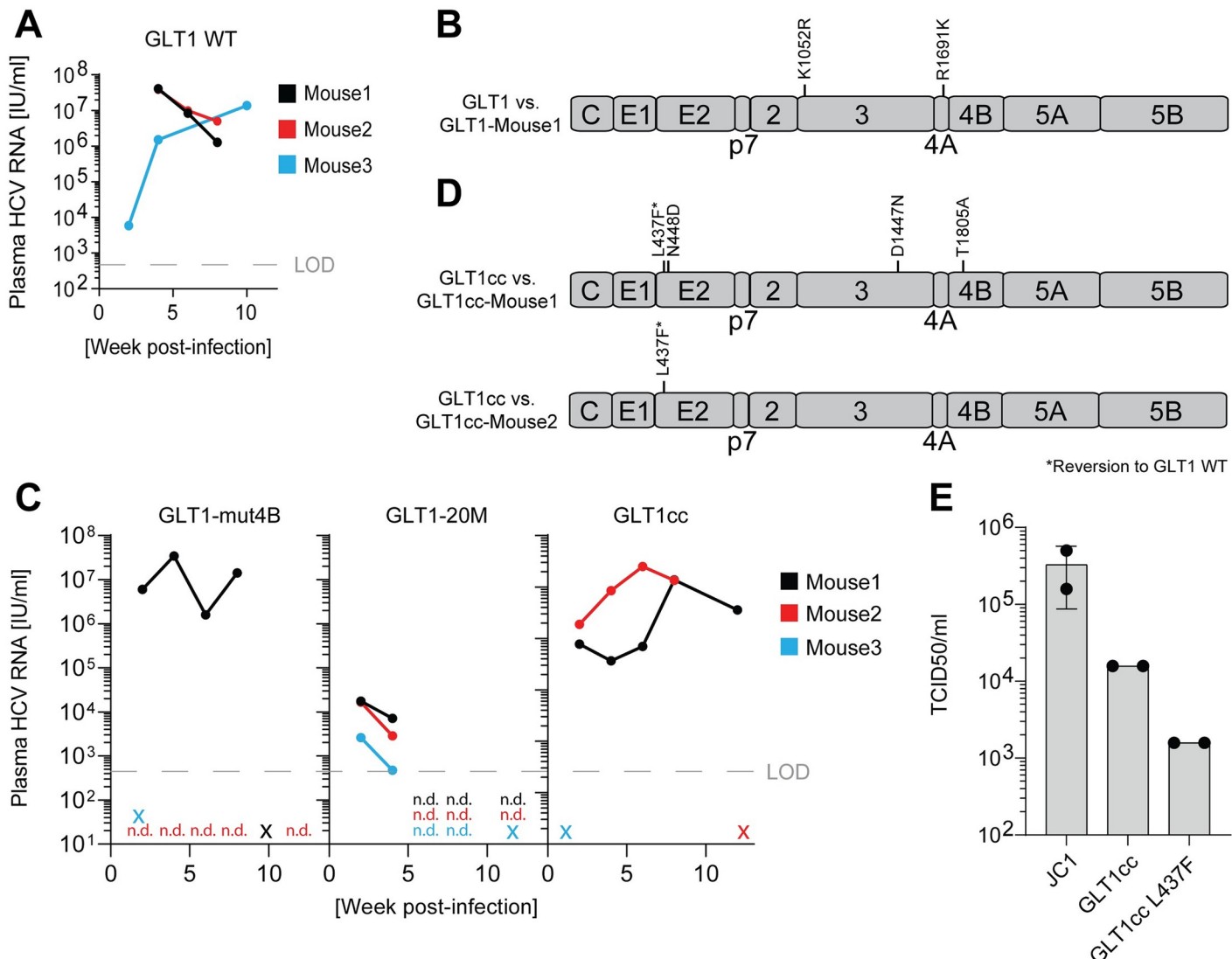

**Fig 7. Infection of human liver chimeric mice.** (**A,C**) Human liver chimeric mice (n = 3) were infected by intrasplenical injection of 50 μl pure high-titer post-transplant serum (A) or they were infected with concentrated supernatants obtained by transfection of GLT1-mut4B, GLT1-20M and GLT1cc (Fig 5) (C). The plasma HCV load was determined at the designated time points using RT-qPCR following total nucleic acid extraction. X = mouse found dead, n.d. = not detectable, LOD = limit of detection. (**B,D**) Consensus amino acid sequences were derived from mouse liver 8 weeks post-infection (B) or from mouse serum after 12 weeks (D, upper panel) or 6 weeks (D, lower panel) and subsequently compared to the respective input consensus sequences. (**E**) Viral titer (TCID50/ml) from concentrated supernatant of indicated HCV isolates harvested from Huh7-Lunet CD81 cells; data from two independent biological replicates.

(Fig 7C, left panel), likely due to the low input titer (see Fig 5). We found no additional conserved mutations upon sequencing of the entire coding region amplified from the serum of mouse 1 at week 6. This result demonstrated that the replication enhancing mutation K1846T did not interfere with infectivity in vivo, as previously demonstrated for the Con1 isolate [19]. In contrast, all three mice infected with GLT1-20M reached only a very transient viremia with titers close to the limit of detection, suggesting that the accumulation of mutations required for titer enhancement in cell culture interfered with replication in vivo (Fig 7C, middle panel). This result was consistent with the lack of reports of replication of any of the highly efficient HCVcc models in vivo, except those based on JFH-1 [29] and gt1a H77S.2 [51], initiating a

low titer transient viremia in human liver chimeric mice. Surprisingly, in case of GLT1cc two mice got productively infected, reaching titers similar to GLT1-mut4B, while again one mouse died early. This result argued for a restoration of in vivo infectivity by the N2415S mutation. Determining the HCV consensus sequence in the serum of both mice revealed a reversion at position 437 back to the GLT1 WT sequence in both cases (Fig 7D). The dominant HCV species in mouse 2 had no further changes compared to GLT1cc, whereas mouse 1 had three additional conserved mutations. This result demonstrated, that GLT1cc indeed is infectious in vivo. Introducing the reversion at position 437 in GLT1cc revealed a negative effect on infectivity in hepatoma cells (Fig 7E). This indicates that the F437L mutation acquired already early upon passaging in cell culture (Fig 3F) is a titer enhancing mutation seemingly having a negative impact in vivo and therefore reverting back to the GLT1 WT sequence.

Taken together, GLT1cc is the first viral isolate of a major genotype capable of efficient replication and infectious virus production in cell culture combined with in vivo infectivity.

## Discussion

In this study we have identified a novel gt1b WT isolate with highly efficient RNA replication in cell culture and developed a new gt1b full-lifecycle cell culture system upon adaptation by serial passaging, generating virions infectious in cell culture and in vivo. Therefore, GLT1 is unique and paradigm breaking in several aspects.

The isolate was identified in a patient serum after two liver transplantations due to its ability to infect and replicate efficiently in hepatoma cells upon PCi treatment [18]. It is tempting to speculate that this clinical background contributes to its outstanding replication capacity, as it was discussed for JFH-1, isolated from an HCV patient with fulminant hepatitis [21] and as it was the case for the gt1a TN strain, again from a fulminant hepatitis C patient, which build the basis for the highly efficient TNcc clone [52]. However, other isolates from post-transplant sera, fulminant or acute hepatitis C did not replicate remarkably well in cell culture or needed a series of adaptive mutations (e.g. JFH-2 [53], BHCV1 [54], NC1 [55]). The ability to analyze replication of WT isolates in regular cell culture has only recently been gained, either by expression of SEC14L2, a lipid transport protein expressed in PHH but absent in Huh7 cells [15] or by PCi treatment, compensating for overexpression of the lipid kinase PI4KA in Huh7 cells, mimicking the mechanism of replication enhancing mutations in NS5A and NS5B [18]. Therefore, systematic studies on multiple WT isolates like JFH-2, BHCV1 or NC1 are still missing but now are in reach. Still, PCi is only efficient for gt1b isolates, since all other genotypes are restricted by additional unknown mechanisms [18]. SEC14L2 in contrast acts pangenotypically in selectable models [15], but has limited stimulatory capacity beyond gt1 for cloned reporter replicons [16]. Nevertheless, the fact that GLT1 and Con1 RNA replication was elevated by both measures to a similar extent, comparable to replication enhancing mutations, suggests that all methods reflect the true replication capacity of an HCV isolate, at least within gt1b. Still, GLT1 further showed a moderately increased replication efficiency in the absence of enhancing conditions, which correlated with a slightly decreased ability to activate PI4KA, compared to Con1. However, this effect was marginal compared to replication enhancing mutations interfering with PI4KA activation [18]. It seems plausible that WT isolates cannot substantially tune down PI4KA activation, due to essential functions of this process in vivo [18–20]. The fact that PCi treatment, acting by the same mechanism, still additively enhanced RNA replication in case of GLT1 underpins the robustness also of this method to reflect the intrinsic replication capacity of a broad range of natural gt1b isolates. Our data therefore hopefully will encourage future studies on the contribution of HCV fitness to persistence and pathogenesis, e.g. by phenotypic analysis of isolates before and after liver

transplantation. It will furthermore be interesting to understand the determinants of the outstanding replication efficiency of GLT1, e.g. by generating intergenotypic chimeras, as in case of JFH-1 [56].

While our understanding of factors limiting HCV RNA replication in cell culture and ways to overcome it has tremendously increased in the last years, it is still technically challenging to generate infectious virus from cloned isolates. Most systems available so far are based on the replicase of gt2a JFH-1, albeit also here the genome has to be adapted to efficient virus production, either by titer enhancing mutations obtained upon passaging or by using the structural proteins of the gt2a isolate J6 (see Ramirez and Bukh for recent comprehensive review [29]). Since replication enhancing mutations have been shown to impair the production of infectious virions to various extent [19,20], the establishment of infectious cell culture models for isolates apart from JFH-1 was very difficult and required tremendous efforts. So far they have been established for gt1a, gt2a, gt2b, gt3a and gt6a, with varying efficiency, requiring up to 20 mutations for efficient virus production (reviewed in [29]), and recently also for gt1b and gt4a [28,31]. Several in vivo infectious gt1b WT isolates have been generated, including Con1 [20], which was the basis for the first efficient RNA replication model [11]. Indeed, by using an assembly neutral replication enhancing mutation in NS4B (mut4B, K1846T), it was possible to show in principle the production of infectious Con1 virus in cell culture and the infectivity of the virions in vivo, using the human liver chimeric mice [19]. However, virus could only be produced in very low amounts early after transfection, suggesting the exclusive packaging of input-RNA; the virus did not spread in culture and could not be further improved upon passaging. Only recently, a more efficient infectious culture model based on Con1 was generated by stepwise adaptation of a chimeric genome harboring NS5B and 3'UTR of JFH-1, requiring a total of 17 mutations [30,31]. In case of GLT1, we used again mut4B and also tried passaging of the WT isolate in SEC14L2 expressing cells, but this time succeeded with mut4B and continuous passaging of cells and supernatants for about one year. It is still not clear, why it was so difficult to generate an efficient gt1b infectious cell culture system, but a striking observation was the fast increase of infected cell numbers after supernatant transfer upon cell passaging, even in absence of detectable amounts of free infectious virus (Fig 3E). Cell-to-cell transmission, which appeared to be a reasonable explanation, is a well-established phenomenon for HCV, defined as infection events that cannot be blocked by neutralizing antibodies [38,57]. We established a co-culture model, allowing image-based analysis of transmission events, to understand whether GLT1 is particularly efficient in cell-to-cell spread, but found no evidence supporting this hypothesis. On the one hand, efficiency of spread in this model correlated with production of infectious virus over the course of serial passaging. On the other hand, the spread of JC1 and DBN3acc, which were used for comparison, was less affected by the presence of neutralizing antibodies than GLT1. These data suggest that cell-to-cell spread generally represents an important transmission mechanism for HCV in cell culture, but still requires the full assembly of infectious virus. However, since all our cell-to cell transmission experiments were based on nuclear translocation of a fluorescent protein from the mitochondrial membrane to the nucleus upon cleavage by the NS3-4A protease, we cannot exclude that increased protease cleavage efficiency among different variants might contribute to apparently higher cell-to-cell transmission in this detection system.

A total of 21 mutations were required to obtain GLT1cc from GLT1 WT. Since we only took three snapshots of the consensus sequence at different stages of passaging, with 6–7 mutations acquired at each step, it is not clear, whether all changes were of functional relevance. Due to the potentially incremental phenotypes expected from individual mutations we refrained from a detailed analysis and instead analyzed the whole replicase and the structural protein coding region separately for changes in RNA replication efficiency and entry/

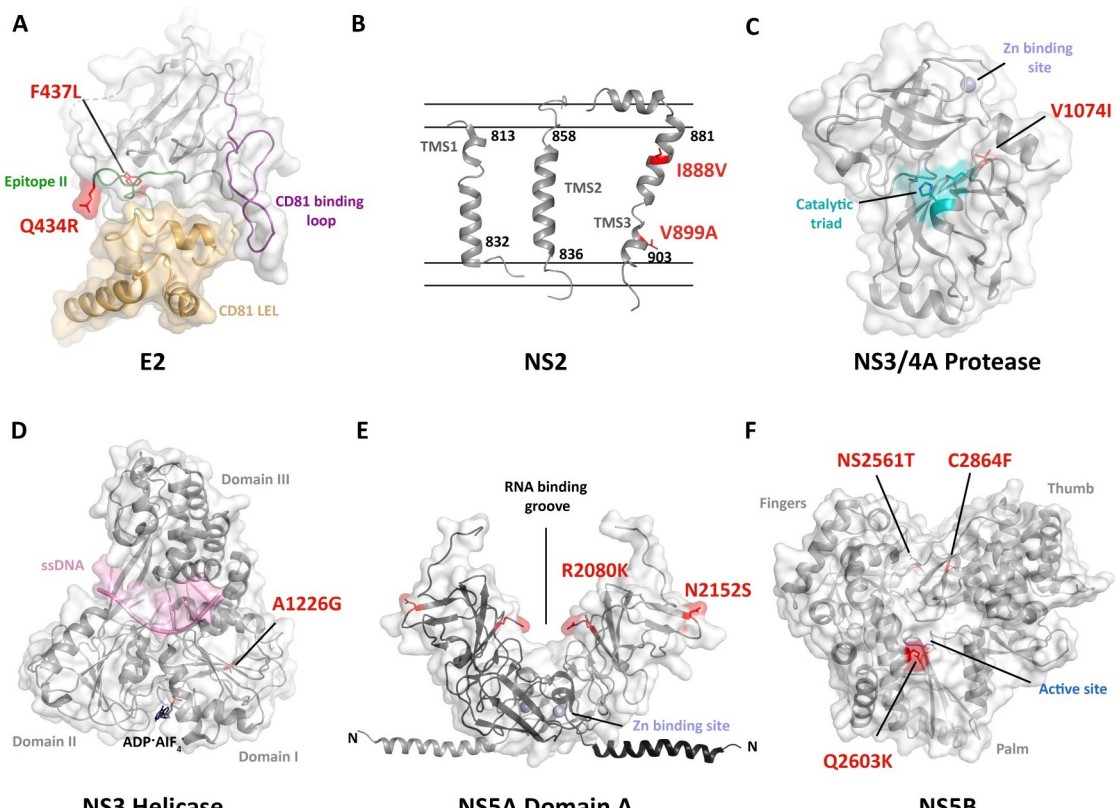

**Fig 8. Localization of the GLT1cc mutations within E2 and non-structural proteins.** The GLT1cc mutations are shown in sticks and highlighted in red. (**A**) Structure of the E2 core (grey) complexed with taumarin CD81 LEL (tCD81 LEL, sand) (PDB: 7MWX). Important E2 antibody binding epitopes named Epitope II and the CD81-binding loop are colored in green and deep purple, respectively. In the crystal structure a histidine is found at position 434 which was mutated *in silico* into a glutamine with the most abundant side chain conformation for clarity. (**B**) Model of NS2 transmembrane segments (TMSs) according to TMS1 (PDB: 2JY0), TMS2 (PDB: 2KWT) and TMS3 (PDB: 2KWZ) are shown as separated entities in cartoon representation. They are tentatively placed into the schematically drawn membrane and the limits of transmembrane helices are given. (**C**) Structure of the NS3/4A N-terminal protease (PDB: 1A1R). Residues of the catalytic triad are shown in sticks and are colored in teal. For clarity, the isoleucine at position 1074 in the crystal structure was mutated *in silico* into a valine with the most abundant side chain conformation. (**D**) NS3 helicase is shown in complex with ssDNA (light pink) and the ATP mimic ADP·AIF4- (black, sticks) (PDB: 3kql). (**E**) The crystal structure of the NS5A Domain 1 from gt1b (PDB: 1ZH1) with the modelled N-terminal amphipathic helix (residues 1–33, PDB 1R7G) and the postulated RNA-binding groove between two monomers. (**F**) NS5B structure (PDB: 2D3U) featuring the characteristic fingers, palm, and thumb domains in light green, yellow and grey, respectively. The aspartic acid residues of the active site are shown in blue sticks for orientation.

assembly, respectively. The substantial stimulation of RNA replication of GLT1-20M/GLT1cc was unexpected given the already high replication capacity of GLT1-mut4B. However, it strengthens our assumption of robust RNA synthesis as a major bottleneck for HCVcc production in Huh7 cells. Although direct contributions to assembly have been reported for all NS proteins, it therefore seems likely that most of the mutations in the NS3-5B region rather contribute to RNA synthesis, except those in domain 3 of NS5A [48]. Mutations in NS3/4A protease/helicase (Fig 8C and 8D) and NS5B polymerase (Fig 8F) are key candidates mediating this phenotype, since they are located within the protein and in part close to the active centers. In case of JFH-1, helicase and polymerase have indeed been shown to underlie the outstanding replication efficiency of this isolate [56] and mutation A1226G was also found in adapted gt1a isolate TNcc [52], as well as upon adaptation of a Con1 5'UTR-NS5A/JFH-1 chimera [30], pointing to a general importance of this alteration in adaptation of gt1. Mutations in the

structured domain 1 of NS5A (Fig 8E) are localized at the surface of the protein and rather do not contribute to the different dimeric isoforms reported (S10 Fig). Their involvement in RNA replication enhancement seems still plausible, but given the multitude of functions of NS5A they might also contribute to assembly [58]. In contrast, domain 3 of NS5A is a key factor in virus assembly, due to its interaction with core [59], that results in the recruitment of NS5A to lipid droplets [60] mediated by CKII dependent phosphorylation of serine residues in this domain [59,61]. In case of GLT1cc, N2415S, located very close to the NS5A-NS5B cleavage site, was particular striking, since it not only had a clear titer enhancing effect in cell culture, but furthermore rescued the low infectivity of GLT1-20M in vivo. A mutation in this region is found in almost all cell culture adapted viruses across genotypes [29]. In case of adapted JFH-1 it was shown that V2440L (corresponding to position 2417 in GLT1) reduced NS5A-NS5B cleavage kinetics and thereby promoted virus assembly [22]. However, the mutation partially reverted to WT in human liver chimeric mice [22], suggesting rather a cell culture specific function. Since N2415S even rescued the in vivo replication capability of GLT1-20M and generated a potential phospho-acceptor site, phosphorylation might be an attractive alternative mechanism [59,61]. However, the fact that the recently established Con1cc model [31] contained a mutation at the same position (D2415G) strengthens the hypothesis that this site might be of general importance for gt1b adaptation, but argues rather against phosphorylation as the underlying mechanism.

NS2 is another important factor of HCV assembly and particularly TMD3, where we found two mutations (Fig 8B), has been identified as a key component [62]. Although this region was not included in the chimeric construct we used to study assembly (C3, [25]), a contribution to virus morphogenesis of GLT1cc appears possible. This assumption is supported by the presence of a nearby mutation in Con1cc (F886L) [31]. Among the other mutations contributing to the increased assembly efficiency we observed for the GLT1cc/C3JFH-1 chimera, F437L is particularly interesting (Fig 8A). On the one hand, it appeared very early in the passaging process and reversion to WT indeed reduced cell culture titers by about 10-fold. On the other hand, it was the only position reverting to WT in both infected human liver chimeric mice, suggesting deleterious effects in vivo. F437L is located within Epitope II, also named AS434, spanning from residue 428 to 446, which is overall highly conserved among HCV genotypes [44,63]. An aromatic residue at position 437 is conserved through all genotypes [63] and distinct mutations implying a reduction of the side chain hydrophobicity at this position reduce binding to CD81 [44], thereby leading to a severe viral entry defect [64] and a reduction in virus fitness [44]. Residue 437 is further part of a structurally conserved 1.5 alpha-helical turn (residues 437–442) in E2 and a mutation in this motif has been proposed to lead to escape from neutralization by HC-84-related antibodies targeting this region [44]. F437L might therefore also be responsible for the reduced neutralizing efficiency of the Fab mix observed for GLT1-20M. The titer enhancing effect we observed for GLT1 might still be due to enhanced virion assembly or release efficiency. Since we used cell lines with ectopic, saturating expression of CD81, we did not observe a reduced entry efficiency of HCVpp. However, in vivo, the defect in CD81 binding seemed to be dominant, likely due to lower receptor expression levels of hepatocytes.

Currently, it appears as a paradigm that virus genomes adapted to cell culture are impaired in vivo. In fact, most available HCV full-length WT genomes are infectious in vivo, either in chimpanzees or in human liver chimeric mice, but do not replicate in cell culture. Unfortunately, limited information is available on the infectivity of their cell culture adapted counterparts [29]. Indeed, initial studies on gt1b Con1 revealed that particularly highly adaptive mutations in NS5A attenuated the isolate in vivo and reverted to WT in chimpanzees [20]. A similar negative correlation and reversions to WT were found in case of gt1a H77S.2, which

initiated a chronic infection in a chimpanzee, but lost all 6 cell culture adaptive mutations during the course of infection [51]. Also for gt1a RTM, a negative correlation between replication in vitro and in human liver chimeric mice was observed [65]. However, these phenotypes are not surprising, since all examples contain replication enhancing mutations in NS5A. Since we now know that these mutations compensate for PI4KA overexpression in Huh7 cells compared to hepatocytes [18], it is plausible, that they are not beneficial to support replication in vivo. A number of comprehensive side-by-side comparisons exists only for JFH-1-based constructs and here the picture is more diverse. JFH-1 titer enhancing mutations have been found to revert to WT upon infection of human liver chimeric mice [22], suggesting again deleterious effects. However, JFH-1 WT had a low capacity for virus production in cell culture and was less pathogenic than other WT strains, requiring also further adaptation in chimpanzees [21,66]. In contrast, J6/JFH-1 chimeras generated much higher virus titers in vitro compared to JFH-1 and replicated more robustly in vivo [45]. Importantly, in case of J6/JFH-1, mutations in NS2 were identified upon infection of human liver chimeric mice stimulating virus titers in vitro and in vivo [67], thereby suggesting that both functions are not necessarily mutually exclusive. Also for gt1b Con1, a replication enhancing mutation in NS4B was found to be compatible with infectivity in vivo [19], therefore we chose it as a starting point for adaptation of GLT1. This might be the reason why GLT1cc, albeit harboring 21 adaptive mutations, kept its ability to replicate in vivo. Interestingly, while GLT1-20M behaved similar to H77S.2 in human liver chimeric mice [51], by initiating only a low level, transient viremia, GLT1cc was rescued by addition of only one critical titer enhancing mutation (N2415S), replicating as robustly as GLT1 WT in vivo, with a critical reversion only at one position in E2 (see above). GLT1cc therefore represents the first highly cell culture adaptive isolate that has retained its infectivity in vivo. By identification of a broader set of mutations facilitating virus production in vivo and in vitro, as in case of N2415S, probably also including A1226G and mutations in TMD3 of NS2, a more rationale based design of additional infectious clones might become in reach.

Given the clinical significance and global abundance of gt1b infections, GLT1cc will represent an important new tool for future vaccine development and studies on HCV pathogenesis, due to its ability to replicate in vitro and in vivo. It is still a long way to regularly culture patient-derived isolates, which would be helpful to understand virus dynamics and heterogeneity and instrumental for vaccine development. However, a better understanding of the molecular mechanisms limiting virion morphogenesis in cell culture will at least allow a more straightforward establishment of infectious culture models based on cloned isolates.

## Materials and methods

### Ethics statement

This study was approved by the ethics committee of the medical faculty of Heidelberg University (ethics vote: S-677/2020). Written consent from the patient was obtained.

### Reagents

The specific PI4KA inhibitor PI4KA-G1 was provided by Glaxo-Smith-Kline [68]; the CKIα-specific inhibitor H479 has been described previously [69]. The PCi treatment was used in a final concentration of 0.1 μM G1 and 5 μM H479 [18]. The HCV neutralizing anti-E2 antibodies AP33 or Fab mix (AR3C-HC84.1 Fab mixture) [70] were used at a concentration of 10 μg/ml. The mouse monoclonal antibody AP33 recognizes conserved residues within the HCV E2 amino acid region 412 to 423 and efficiently neutralizes diverse HCV genotypes in cultured cells [41,42]. The monoclonal IgG2α mouse antibody 9E10 was used for detection of NS5A by

immunofluorescence (dilution 1:1,000). PI4P was visualized using a monoclonal IgM mouse antibody (Z-P004; Echelon) with a dilution of 1:200. As secondary antibody for immunofluorescence, anti-mouse IgG/IgM AlexaFluor 488 and anti-mouse IgG/IgG2α AlexaFluor 546 (both Invitrogen) were used. To detect SEC14L2 in western blot, a monoclonal rabbit antibody (Abcam) was used at a dilution of 1:5,000. As secondary antibody for western blot, a goat anti-rabbit HRP antibody (Sigma-Aldrich) was used.

## Plasmid constructs

To generate cell lines stably expressing SEC14L2 the previously described pWPI-BLR SEC14L2 (isoform 1) construct was used [16].

For replication studies, pFK i341 PiLuc NS3-3´ Con1 WT, pFK i341 PiLuc NS3-3´ Con1 K1846T (mut4B), pFK i341 PiLuc NS3-3´ Con1 S2204R (mut5A), pFK i341 PiLuc NS3-3´ Con1 R2884G (mut5B) [17] and pFK i341 PiLuc NS3-3´ JFH-1 WT [71] have been described previously. pFK i341 PiLuc NS3-3´ GLT1 K1846T (mut4B), pFK i341 PiLuc NS3-3´ GLT1 S2204R (mut5A) and pFK i341 PiLuc NS3-3´ GLT1 R2884G (mut5B) were generated with oligonucleotides as specified in S1 Table. Subgenomic replicons of GLT1-20M and GLT1cc were constructed with the NEBuilder HiFi DNA assembly cloning kit (New England Biolabs) according to the manufacturer's instructions using oligonucleotides as specified in S1 Table.

For CLEM experiments, the pTM NS3-5B Con1-NS5A_mCherry and pTM NS3-5B GLT1-NS5A_mCherry constructs with an in-frame insertion of mCherry in domain 3 of NS5A (placement according to previous descriptions [72]) were generated. pTM vectors allow for transient protein expression of HCV non-structural proteins under control of the T7 promotor.

For infection experiments, plasmids encoding full-length genomes of JC1 [25], Con1 WT [11] and DBN3acc [40] have been described elsewhere. pFK GLT1-mut4B, pFK GLT1-mut4B +N2415S, pFK GLT1-20M+N2415S (GLT1cc), pFK Con1/C3GLT1 and pFK GLT1cc L437F were generated with oligonucleotides as specified in S1 Table.

For studies on particle production, pFK i389 JCN2A and pFK i389 JCN2AΔGDD have been described elsewhere [47]. pFK i389 JCN2AΔE1E2, pFK i389 GLT1/C3JFH-1 N2A and pFK i389 GLT1cc/C3JFH-1 N2A were created with oligonucleotides as specified in S1 Table.

Regarding the HCVpp, pCMVΔ8.74 [73], pHR'CMVLuc [39] and pcDNAΔcE1E2-Con1 [72] were described previously. To create the expression vectors for the HCVpp of GLT1 and GLT1cc, oligonucleotides were used as specified in S1 Table.

## Cell lines

All eukaryotic cells were cultured at 37˚C in a constant humid atmosphere containing 5% (v/v) $CO_2$. Cells were cultured in plastic dishes or flasks in Dulbecco's modified Eagle's medium (DMEM; Gibco) containing 10% (v/v) FCS (Seromed; inactivated for 30 min at 56˚C), 1% (v/v) penicillin/streptomycin (10,000 U/ml penicillin, 10,000 μg/ml streptomycin; Gibco), 2 mM L-glutamine (Gibco) and 1% (v/v) 100× non-essential amino acids (Gibco). The identity of all basic cell lines (Huh7 High Passage, Huh7-Lunet and Huh7.5) was verified by a Multiplex human cell line authentication test. All cell lines were regularly tested to check they were free of mycoplasma contamination using a commercially available system (MycoAlert Mycoplasma Detection kit; Lonza). HCV replication experiments were conducted using the highly permissive human hepatoma cell line Huh7 High Passage [17] or the sub-clones Huh7-Lunet [18] or Huh7.5 [33]. Huh7-Lunet T7 stably express the T7 RNA polymerase [74]. Ectopic CD81 expression is required due to low expression in Huh7-Lunet [39]. Huh7-Lunet CD81N, selected for the absence of CD81, have been described elsewhere [38]. Huh7-Lunet CD81 cells

stably expressing MAVS–GFP–NLS [36] have been described [18]. Expression of MAVS–GFP–NLS or MAVS-mCherry-NLS allowed detection of HCV replication in individual live cells. The construct encodes enhanced GFP or mCherry respectively with an NLS fused to the carboxy-terminus of MAVS, containing an HCV NS3/4A cleavage site and tagging the protein to mitochondria. Huh7-Lunet; Huh7 and Huh7.5 cells expressing SEC14L2 were generated with lentiviral vectors based on plasmid pWPI-BLR SEC14L2 (isoform 1) [16], expression was maintained by selection with blasticidin (5 μg/ml).

## Patient information

Serum for isolation of GLT1 was obtained from a male patient under immunosuppression with tacrolimus, mycophenolic acid and methylprednisolone because of status post re-liver-transplantation. For chronic HCV infection, the patient had been treated with pegylated interferon α/ribavirin 10 years before and with pegylated interferon α/ribavirin/sofosbuvir 1 year before. Note that the same serum sample was used in our previous study and referred to as "Serum 1" in Fig 5G and 5H of the respective publication [18].

## RNA extraction

RNA was extracted from hepatoma cells using the NucleoSpin RNA plus kit (Macherey-Nagel) according to the manufacturer's instructions.

GLT1 infected human liver chimeric mice were sacrificed at 8 weeks post infection. After collection of the livers, 100 mg of tissue samples were preserved frozen in 1.5 ml RNAlater solution (ThermoFisher Scientific). To extract RNA, 20 mg of tissue were grinded to powder using a Dounce tissue grinder set (Merck) on dry ice. RNA was then isolated using the Bio&SELL RNA-Mini Kit (Bio&SELL) according to the manufacturer's protocol.

RNA was extracted from the human GLT1 patient serum as well as from the serum of GLT1-mut4B, -20M and GLT1cc infected human liver chimeric mice using the NucleoSpin Virus kit (Macherey-Nagel) according to the manufacturer's instructions.

## Sequence analysis of viral populations

GLT1 WT sequence was obtained from total RNA extracted from Huh7-Lunet CD81 cells infected with Serum 1 and treated with PCi [18]. cDNA synthesis and nested PCR amplification of the HCV coding sequence and parts of the noncoding sequence was done in three overlapping fragments with the Expand-RT and Expand-long-PCR system (Roche) using primers and protocols as described for Con1 (S1 Table, [11]). By direct sequencing of PCR products, a consensus sequence was obtained and synthesized (GeneCust Europe). The termini of the genome (1–82 and 3' half of the x-tail) were taken from the Con1 sequence.

For analysis of the GLT1 cell culture adaptation as well as for the HCV population in infected mice, isolated RNA was converted into cDNA using the SuperScript IV First-Strand Synthesis kit (Invitrogen) with the HCV specific 3´UTR antisense primer A_9416 (S1 Table) according to the manufacturer's instructions. From the cDNA, the HCV genome was amplified in three overlapping fragments via nested PCR using the platinum SuperFi II DNA polymerase kit (Invitrogen) with oligonucleotides as specified in S1 Table. After the nested PCR, PCR products were purified via agarose gel electrophoresis and directly sequenced. In case of GLT1-20M, the consensus of the coding sequence was synthesized (GeneArt) and inserted into the GLT1 WT construct.

In order to extract individual viral clones from the pooled PCR samples derived from the cDNA, the TOPO XL-2 Complete PCR Cloning Kit (Invitrogen) was used according to the manufacturer's instructions.

## In vitro transcription of HCV RNA, electroporation of cells and luciferase activity assay

In vitro transcription, electroporation and luciferase assays were performed as described elsewhere [17]. In brief, 2.5–5 µg of in vitro transcribed viral RNA was electroporated into 2-4x10$^6$ cells for Huh7-Lunet or Huh7 High Passage cells, or 3-6x10$^6$ cells for Huh7.5 cells. The cells were seeded into 6- or 12-well plates and treated with DMSO or PCi (5 µM H479 / 0.01 µM G1) [18] after four hours. Afterwards, the cells were harvested at the indicated time points, and luciferase activity was determined in duplicates using a tube luminometer (Lumat LB9507; Berthold Technologies). If applicable, luciferase activity at 4 h after transfection was used for normalization to account for varying transfection efficiency. Nano luciferase measurements were performed with the Nano-Glo Luciferase Assay System (Promega) according to the manufacturer's instructions.

## Low precision correlative light and electron microscopy (low-precision CLEM)

For the analysis of the replication organelles in the replication competent system, 4x10$^4$ electroporated Huh7-Lunet CD81 MAVS-GFP-NLS cells with or without SEC14L2 expression were seeded in a 6 cm dish with a grid on the bottom (MatTek Corporation) and were incubated at 37˚C for 48 h. In case of the expression model, 4x10$^4$ Huh7-Lunet T7 cells with or without SEC14L2 expression were seeded in a 6 cm-diameter-dish with a grid on the bottom (MatTek Corporation) and were transfected 24 h later with pTM NS3-5B Con1 NS5A_m-Cherry or pTM NS3-5B GLT1 NS5A_mCherry. Prior to the fixation either the nuclear GFP localization or the NS5A_mCherry signal was used to determine regions on the grid with a high number of HCV positive cells. Afterwards, cells were fixed with 2.5% GA, 2% sucrose in 50 mM sodium cacodylate buffer (CaCo), supplemented with 50 mM KCl, 2.6 mM MgCl$_2$ and 2.6 mM CaCl$_2$ for at least 30 min on ice. After three washes with 50 mM CaCo, samples were incubated with 2% osmium tetroxide in 25 mM CaCo for 40 min on ice, washed three times with EM-grade water and incubated in 0.5% uranyl acetate in water overnight at 4˚C. Samples were rinsed three times with water, dehydrated in a graded ethanol series (from 40% to 100%) at RT, embedded in Epon 812 (Electron Microscopy Sciences) and polymerized for at least 48 h at 60˚C. After polymerization, the negative imprint of the coordinate system from the gridded coverslip was used to identify the areas of interest for ultrathin sections of 70 nm by sectioning with an ultramicrotome Leica EM UC6 (Leica Microsystems) and were afterwards mounted on a slot grid. Sections were counterstained using 3% uranyl acetate in 70% methanol for 5 min and lead citrate (Reynold's) for 2 min and imaged by using a JEOL JEM-1400 (JEOL) operating at 80 kV and equipped with a 4K TemCam F416 (Tietz Video and Image Processing Systems).

## Immunofluorescence

Specific staining of PI4P on intracellular membranes has been described elsewhere [75]. In brief, for expression of the HCV NS3-5B proteins, 4×10$^4$ Huh7-Lunet T7 cells were seeded 24 h prior to transfection in a 24-well plate on coverslips. Transfection was done with LT1 transfection reagent (Mirus Bio LLC, Madison, WI, USA) according to the manufacturer's instructions. Cells were fixed 18 h post-transfection in 4% PFA for 20 min at RT followed by permeabilization with 0.5% Digitonin-PBS for 20 min. The permeabilized cells were then blocked for 1 h in 5% BSA-PBS prior to incubation with the primary antibodies in 3% BSA-PBS for 1 h at RT. After three washing steps with PBS each for 5 min, the secondary

antibodies (Alexa Flour, Invitrogen) were incubated in 3% BSA–PBS for 45 min in the dark at RT. After three washing steps with PBS each for 5 min, nuclei were stained with a 1:4,000 TBS dilution of 4′,6-diamidino-2-phenylindole (DAPI) for 1 min. Cells were mounted with Fluoromount G (Southern Biotechnology Associates), and images were acquired with a Leica SP8 confocal laser-scanning microscope (Leica Microsystems). Quantification of cellular PI4P levels was performed as previously described [18]. Briefly, whole-cell z-stacks of the PI4P channel were recorded as 8-bit TIFF files using 20 slices of around 0.4 μm thickness with a ×40 (numerical aperture (NA) 1.4) objective of at least three different randomly chosen fields of view. For each stack, a single slice was recorded for the NS5A channel to identify NS5A-positive cells for PI4P quantification. For quantification of NS5A signals, stacks were recorded in an identical manner as for PI4P. Image stacks were z-projected according to their maximum intensity in Fiji and a constant threshold was set to 30–40, creating a binary image. The NS5A channel was overlaid, NS5A-positive cells were encircled and integrated density values of the thresholded PI4P signal were acquired by Fiji software.

## Production of cell culture derived virus

Production of virus was performed as described elsewhere [19]. Briefly, full-length RNA genomes were electroporated into Huh7-Lunet or Huh7.5 cells. Supernatants were collected at 24 and 48 hours after transfection and filtered through 0.45 μm filter units. Afterwards, the collected supernatants of each isolate were concentrated using a Centricon Plus-70 centrifugal filter device (100 K nominal molecular weight limit; Millipore) resulting in an approximately 100x concentrated stock (v/v) that was aliquoted and stored at -80˚C.

## Core-ELISA

For the determination of extracellular core protein level, 900 μl supernatant was filtered (0.45 μm filter) and 100 μl 10% Triton X100-PBS was added to a final concentration of 1% Triton X100. Intracellular core protein was determined by resuspension of ~$1x10^6$ cells in 1 ml 1% Triton X100-PBS. Core protein was quantified using a commercial chemiluminescent microparticle immunoassay (ARCHITECT HCV Ag Reagent kit; Abbot Diagnostics) according to the manufacturer's instructions. Samples were analyzed by the central laboratory for diagnostics at the University Hospital Heidelberg.

## TCID50

To assess the tissue culture infectious dose (TCID50) of virus containing supernatant or virus stock, we seeded Huh7-Lunet CD81 MAVS-GFP-NLS cells at a concentration of $1x10^4$/well in a 96-well plate 24 h prior to the infection. The medium was aspirated and 200 μl of a 1:10 diluted inoculum was added to the first row, followed by a subsequent 1:10 dilution in each row. After 4 h, the inoculum was removed, fresh DMEM medium was added to the cells and incubated for 72 h at 37˚C with 5% $CO_2$. Cells were washed once with PBS and fixed with 4% PFA for 20 min at RT, followed by two wash steps with PBS. The infection status of each individual well was determined by fluorescent microscopy based on the location of the GFP signal. The TCID50 was calculated using the method of Spearman and Kärber as described previously [76].

## Real-time quantification PCR (Taqman qPCR)

HCV RNA copies were determined by Taqman RT-qPCR using the Quanta BioSciences qScript XLT One-Step RT-qPCR kit according to the manufacturer's instructions (Bio-Rad).

GLT1_probe was used to detect for GLT1 RNA, together with the S_59 & A_165 primers (S1 Table). For detection of JC1 RNA, JC1_probe was used together with the S_146 & A_219 primers (S1 Table). Taqman RT-qPCRs were run in triplicate together with an HCV RNA standard for each isolate of known quantity and analyzed using Bio-Rad CFX96 software.

## Passaging of cells and supernatants

5 µg of in vitro transcribed viral full-length RNA was electroporated into $4x10^6$ Huh7-Lunet CD81 cells either expressing SEC14L2 or not. The cells were seeded in a 6-well dish 1:1 together with non-electroporated cells expressing additionally a MAVS-GFP-NLS reporter protein. After 72–96 h, the cells were expanded/passaged in a 1:5 dilution. At the indicated time points, the supernatant (~60 ml) of three 15 cm-diameter dishes was concentrated (~600 µl) and used for infection of previously seeded (12-well) $4x10^4$ Huh7-Lunet CD81 MAVS-GFP-NLS cells with or without SEC14L2 expression. 4 hours post infection, the inoculum was removed, replaced with 1 ml fresh DMEM medium and incubated for 72 h at 37°C with 5% $CO_2$. The putative infected cells were fixed with 4% PFA and manually analyzed for nuclear GFP signal. In case of GLT1-mut4B p29 and all the following supernatant passages, the infected cells, instead of the initially electroporated cells, were expanded/passaged each 72–96 h as described above. After the indicated passage number, supernatant (~7 ml) from a 10 cm-diameter-dish was used to infect previously seeded (10-cm-diameter-dish) $1x10^5$ Huh7-Lunet CD81 MAVS-GFP-NLS for 4 hours. Afterwards the inoculum was removed, replaced with 8 ml fresh DMEM medium and incubated for 72–96 h. Prior to each cell passaging and/or supernatant transfer, the number of nuclear GFP signals within the culture was estimated via fluorescence microscopy.

## Production and purification of recombinant Fab fragments

Fabs from antibodies HC84.1 and AR3C were cloned and produced as described previously [70]. Briefly, codon-optimized synthetic genes (Genscript) encoding heavy and light chains of the Fab regions were cloned into a bicistronic Drosophila S2 Fab expression vector comprising a double Strep tag at the C-terminus of the heavy chain and an N-terminal BiP-signal sequence for efficient translocation of each chain. Next, stably transfected S2 cell lines were induced with 4 µM $CdCl_2$ at a density of approximately $6x10^6$ cells/ml for Fab production. After 6 days, Fabs were purified from the cell supernatant using affinity chromatography (Strep Tactin XT Superflow resin, IBA) followed by size exclusion chromatography (SEC; HiLoad 26/600 Superdex 200 pg column, GE Healthcare). Both proteins were mixed in a molecular ratio 1:1 and the AR3C-HC84.1 Fab mixture was re-purified by SEC on a Superdex 200 Increase 10/300 GL column (GE Healthcare) with PBS as the eluent.

## Viral spread assay

HCV positive Huh7-Lunet CD81 MAVS-GFP-NLS, either electroporated or passaged cells, were co-seeded in a 12-well dish with naïve Huh7-Lunet CD81 MAVS-mCherry-NLS in a ratio of 1:5. The cells were harvested at indicated time points with 4% PFA-PBS for 20 min at RT. Each well was additionally stained with DAPI. Afterwards, the whole well was imaged using a Celldiscoverer 7 microscope using an Axiocam 712 and a Plan-Apochromat 5x/0.35 objective (all Carl Zeiss Microscopy).

## Quantification of nuclear GFP or mCherry in Huh7-Lunet CD81 cells

To segment nuclei in stitched images we trained a Random Forest classifier based on the DAPI signal using iLastik [77] which predicts semantic class attribution (nucleus or background) for

every pixel. Objects obtained in this way were subsequently filtered by size to exclude unusually small or large nuclei which often represent aberrant biological structures or microscopy artefacts. Hysteresis algorithm was used to separate nuclei in close proximity to each other. In the next step, using iLastik object classification workflow, a machine learning algorithm was trained to classify objects into three categories—infected, non-infected and HCV spreading "seed" cells based on the localization of the infection reporters in GFP and mCherry channel. The training set of data was arbitrary selected, and the same machine learning algorithm was used for the pixel and object classification in all images.

## Quantification of cell distribution and foci of infected cells

Based on the automated image analysis and cell classification by iLastik, the spatial distribution of donor and recipient cells, as well as quantification of individual foci of infected cells was evaluated as follows. Distances between two individual cells were calculated based on the Euclidean distance, i.e., $d = \sqrt{(x_1 - x_2)^2 + (y_1 - y_2)^2}$ with $(x_1, y_1)$ and $(x_2, y_2)$ defining the positions of the nuclei of the cells as identified by iLastik. As image analysis does not allow visualization of contacts of individual cell membranes, we determined the cell composition within the local surrounding of a donor cell by considering all cells with nuclei positions within a certain radius $r$ around the cell. With individual cells assumed to have a diameter of 20–30 μm, we set $r = 50$ μm, but also tested other radii (30 μm, 100 μm) which did not change our results. Cellular composition was then assessed by the cell type classifications of all cells within this area as obtained from iLastik. Furthermore, we determined the minimal distance of each donor cell to the nearest recipient cell by calculating the distances between each of the different cell types and sorting them accordingly. Foci of infected cells were determined by considering each cell within a certain distance $d_n$ to be in contact to the other cell. Individual foci and their sizes were then assessed by counting all connected donor and recipient cells. We defined donor foci as those that contained at least one donor cell, and growing donor foci comprising at least one donor and recipient cell each. As before, $d_n$ was set to 50 μm to define connected cells, with the use of other values ($d_n = 30$ μm) not changing our results.

## HCVpp

Human Immunodeficiency virus (HIV)-based particles bearing HCV envelope proteins were produced by HEK293T cells. $1.2x10^6$ cells were seeded in a 6 cm-diameter-dish and transfected with 2.16 μg envelope protein expression construct pcDNAΔcE1E2-Con1 [72], pcDNAΔcE1E2-GLT1 or pcDNAΔcE1E2-GLT1cc, 6.42 μg HIV gag-pol expression construct pCMVΔ8.74 [73] and 6.42 μg firefly luciferase transducing retroviral vector [78] using polyethylenimine (PEI). Medium was replaced after 6 h. After 48 h, supernatant containing the pseudoparticles was passed through a 0.45 μm filter and used to infect $4x10^4$ naïve Huh7-Lunet CD81 cells seeded in a 12-well plate the day before. After 72 h, luciferase assay was performed. SYBR Green based Product Enhanced Reverse Transcriptase assay (SG-PERT) was performed using the Takyon SYBR green kit (Eurogentec) to quantify HCVpp titers used for infection [79].

## Sequence analysis

358 full-length HCV gt1b polyprotein sequences were obtained from the NIAID Virus Pathogen Database and Analysis Resource (ViPR) [80] through the web site at http://www.viprbrc.org/. With these sequences, a phylogenetic tree based on the minimum evolution principle [81] was constructed and visualised with FigTree (v1.4.4). The gt1b consensus sequence was

generated with the HCV sequences used for the phylogenetic tree; sequences were aligned with Clustal Omega and the consensus sequence was derived with the EMBOSS Cons tool, both from the EMBL-EBI sequence analysis tools [82]. The dataset underlying the gt1b consensus was further used to determine the frequency of the mutations arising in GLT1cc with the help of the Metadata-driven Comparative Analysis Tool (meta-CATS) of ViPR [83].

## Mice

Human liver chimeric mice were generated by transplantation of approximately $10^6$ primary human hepatocytes (donor L191501 from Lonza, Switzerland) into homozygous uPA$^{+/+}$-SCID mice as previously described [84]. Human albumin quantification in mouse plasma was used to assess the level of liver humanization. Mice (n = 3 per group) were infected by intrasplenical injection of the respective viral inoculum (GLT1-mut4B, GLT1-20M and GLT1cc). Blood plasma was collected at a two-weekly base and the plasma HCV load was determined by Real-Star HCV RT-qPCR (Altona Diagnostics) following total nucleic acid extraction (NucliSENS EasyMag, BioMérieux).

## Software

Fluorescence, western blot and electron microscopy images were analyzed using Fiji. Sequence alignments were visualized with AlignX, Sanger sequencing results were analyzed and visualized with ContigExpress both of the Vector NTI software package (Life Technologies). Analysis of the spatial distribution of cells was performed in R (https://cran.r-project.org). Figures were arranged with Adobe Photoshop and Adobe Illustrator. Schematics were created with BioRender.com.

## Statistical analysis

Statistical analyses were performed using GraphPad Prism 8. Unless otherwise indicated, statistics of data following a normal distribution and having similar variance were calculated using an unpaired two-tailed Student's t-test.

## Supporting information

**S1 Fig. SEC14L2 expression, schematic of the experimental procedure and replication efficiency of GLT1 in Huh7-Lunet cells.** (**A**) SEC14L2 expression levels in Huh7-Lunet, Huh7.5 and Huh7 cells after lentiviral transduction with a SEC14L2 encoding vector or empty control, compared to primary human hepatocytes (PHH). Approximately 1x10$^5$ Huh7-Lunet, Huh7.5, Huh7 or PHH were lysed and analyzed by 10% SDS-PAGE/Western blotting for SEC14L2 and Calnexin expression. One representative experiment from two independent repetitions is shown. (**B**) Schematic of a subgenomic reporter replicon and of the experimental procedure of luciferase-based replication measurement. HCV 5'UTR (HCV), poliovirus internal ribosomal entry site (IRES) (PV) and EMCV-IRES (EMCV), as well as the HCV 3'UTR are indicated by their respective secondary structures. NS3-NS5B and firefly luciferase coding regions are indicated by orange and yellow boxes, respectively. Transfection by electroporation is visualized by a lightning-symbol. (**C**) Replication enhancement by a combination of mutations and SEC14L2 expression for Con1 (left) and GLT1 (right). Huh7-Lunet cells either expressing SEC14L2 or transduced with empty vector were electroporated with the indicated subgenomic reporter replicon RNA containing mutations in NS4B (K1846T), NS5A (S2204R) or NS5B (R2884G) as indicated. Luciferase activity in cell lysates (RLU) was quantified as a correlate of RNA replication efficiency at the given time points and normalized to 4 h. A replication

deficient Con1 variant (Con1ΔGDD) was used as a negative control and the respective luciferase level at 72 h is indicated by a dashed grey line in all diagrams. The data are the mean values from two independent experiments shown as individual data points with two technical replicates each.
(TIF)

**S2 Fig. Replication efficiency of GLT1 compared to Con1 in Huh7 cells, using different replication enhancing conditions.** Huh7 cells were transfected with subgenomic reporter replicons of the indicated isolates or mutants. Luciferase activity in cell lysates (RLU) was quantified as a correlate of RNA replication efficiency at the given time points and normalized to 4 h. (**A,C**) HCV replication was stimulated by SEC14L2 expression compared to empty vector transduction as indicated. (**B,C**) Replication enhancement of GLT1 (green lines) or Con1 (black lines) by mutations in NS4B (K1846T), NS5A (S2204R) or NS5B (R2884G). A replication deficient Con1 variant (Con1ΔGDD) was used as a negative control for replication and the respective luciferase level at 72 h is indicated by a dashed grey line in all diagrams. The data are the mean values from two independent experiments shown as individual data points with two technical replicates each.
(TIF)

**S3 Fig. Replication efficiency of GLT1 compared to Con1 in Huh7.5 cells, using different replication enhancing conditions.** Huh7.5 cells were transfected with subgenomic reporter replicons of the indicated isolates or mutants. Luciferase activity in cell lysates (RLU) was quantified as a correlate of RNA replication efficiency at the given time points and normalized to 4 h. (**A,C**) HCV replication was stimulated by SEC14L2 expression compared to empty vector transduction as indicated. (**B,C**) Replication enhancement of GLT1 (green lines) or Con1 (black lines) by mutations in NS4B (K1846T), NS5A (S2204R) or NS5B (R2884G). A replication deficient Con1 variant (Con1ΔGDD) was used as a negative control for replication and the respective luciferase level at 72 h is indicated by a dashed grey line in all diagrams. The data are the mean values from two independent experiments shown as individual data points with two technical replicates each.
(TIF)

**S4 Fig. Ultrastructural analysis of membrane rearrangements after expression of GLT1 NS3-5B NS5A_mCherry.** Huh7-Lunet T7 cells either expressing SEC14L2 (T7 SEC14L2) or not (T7 empty) were transfected with a pTM vector encoding either Con1 or GLT1 NS3-5B with an in frame insertion of mCherry in domain 3 of NS5A [72]. Cells were fixed 24 hours post transfection and the mCherry fluorescent signal was used to identify positive cells for further analysis by CLEM. (A) A minimum of six cells were analyzed and the DMV diameter was measured. Unpaired two-tailed Student's t-test was used to determine statistical significance (**** = p ≤ 0.0001). (B) Representative images of each condition.
(TIF)

**S5 Fig. Virus production in Huh7.5 cells and analysis of a chimeric GLT1 genome harboring the structural proteins of Con1.** (**A**) Examples of lack of infection events of indicated isolates upon supernatant transfer to MAVS-GFP-NLS cells. Infection is identified by nuclear GFP signal. (**B**) Detection of intra- and extracellular core protein after transfection of full-length virus genomes. Huh7.5 cells with or without SEC14L2 expression were transfected with the indicated HCV full-length genomes and intra- and extracellular core protein levels were determined by ELISA as correlates of replication and virus secretion, respectively. (**C,D**) A chimeric GLT1 genome, encoding the structural proteins of Con1 up to the C3 junction site in

NS2 [25] was transfected in Huh7-Lunet (C) or Huh7-Lunet CD81 (D) cells with or without SEC14L2 expression and/or PCi treatment. Intra- and extracellular core levels were determined by Elisa 72 hours post transfection (C) or before passaging (D). Shown are data from two independent experiments (B, C) or from a single passaging experiment (D). The dashed grey line indicates the detection limit of 3 fmol/l core protein; n.d. = not detectable.
(TIF)

**S6 Fig. Input viral titers and analysis of spreading properties of GLT1-20M compared to JC1.** (**A**) Viral titer (TCID50/ml) from concentrated supernatant of different HCV isolates harvested from Huh7-Lunet CD81 MAVS-GFP-NLS used in Fig 4D in the mock condition. Shown are the mean values and standard deviation of two independent experiments shown as individual data points. (**B,C**) Distribution of the shortest distances between a GLT1-20M (B) or JC1 (C) positive Huh7-Lunet CD81 MAVS-GFP-NLS cell ("donor") and the next nearest Huh7-Lunet CD81 MAVS-mCherry-NLS with nuclear mCherry signal ("recipient") after treatment with HCV neutralizing antibodies or left untreated (mock). Numbers in brackets and vertical lines indicate the median distance for each treatment. Shown are representative data from one experiment (n = 2).
(TIF)

**S7 Fig. Representative still images from live cell imaging of virus spread assays.** Every 30 minutes an image was acquired for 72 hours. Donor cells of interest are labelled with a cyan arrow, recipient cells of interest are labelled with a white arrow. (**A**) Example of cell-to-cell mediated spread in both GLT1cc (S1 Movie) and JC1 (S2 Movie). (**B-E**) Examples based on JC1 because of the overall higher number of infection events. (**B**) Infection in absence of a visible donor cell representing cell-free spread (S3 Movie). (**C**) Cell-to-cell spread with recipient cell moving away from donor cell (S4 Movie). (**D**) Vanishing of donor cell after cell-to-cell spread (S5 Movie). (**E**) Earliest observed infection event.
(TIF)

**S8 Fig. Identification of polymorphic sequence N2415S in the quasispecies of p118.27 and titer of GLT1cc in Huh7.5 cells.** (**A**) Sequences of 5 individual subclones of RT-PCR products obtained from total RNA of p118.27 compared to GLT1-20M. A missing consensus mutation at position 1074 is shown in blue, premature stop codons are underlined and highlighted in bold. N2415S is shown in red color. (**B**) Sequence chromatogram of the region encoding N2415S obtained from total RNA of p118.27 by direct sequencing of the amplicon. (**C**) Viral titer (TCID50/ml) from concentrated supernatant of indicated HCV isolates harvested from Huh7.5 cells; data from two independent biological replicates.
(TIF)

**S9 Fig. Sequence chromatograms of individual base positions with a shift in abundance in the GLT1-patient serum compared to the mouse serum 8 weeks after GLT1 WT infection.**
(TIF)

**S10 Fig. Alternative NS5A domain 1 dimer structures.** The NS5A domain 1 is anchored to phospholipid membranes by an N-terminal amphipathic helix (residues 1–33, PDB 1R7G). Crystal studies revealed four different dimeric forms of domain 1 from genotype 1a and 1b with the same monomeric unit, but different dimeric arrangements shown. (**A**) The first crystal structure from genotype 1b (PDB: 1ZH1) with the modelled N-terminal amphipathic helix and the postulated RNA-binding groove between the two monomers is shown. (**B**) Monomers from genotype 1a (PDB: 4CL1) form a head to head dimer. (**C**) Genotype 1b monomers assemble in parallel to form an extensive interface (PDB: 3FQM). (**D**) Monomers from

genotype 1a dimerize via a similar interface as shown in C but are assembled in an antiparallel fashion (PDB: 4CL1).
(TIF)

**S1 Text. Supplementary methods.**
(DOCX)

**S1 Table. DNA oligonucleotides used in this study.**
(DOCX)

**S1 Movie. Example of cell-to-cell mediated spread in GLT1cc.** Live cell imaging of cell spread assay. Every 30 minutes an image was acquired for 72 hours. Donor cells were labelled with GFP and recipient cells with mCherry.
(MP4)

**S2 Movie. Example of cell-to-cell mediated spread in JC1.** Live cell imaging of cell spread assay. Every 30 minutes an image was acquired for 72 hours. Donor cells were labelled with GFP and recipient cells with mCherry.
(MP4)

**S3 Movie. Infection in absence of a visible donor cell representing cell-free spread in JC1.** Live cell imaging of cell spread assay. Every 30 minutes an image was acquired for 72 hours. Donor cells were labelled with GFP and recipient cells with mCherry.
(MP4)

**S4 Movie. Cell-to-cell spread with acceptor cell moving away from donor cell in JC1.** Live cell imaging of cell spread assay. Every 30 minutes an image was acquired for 72 hours. Donor cells were labelled with GFP and recipient cells with mCherry.
(MP4)

**S5 Movie. Vanishing of donor cell after cell-to-cell spread in JC1.** Live cell imaging of cell spread assay. Every 30 minutes an image was acquired for 72 hours. Donor cells were labelled with GFP and recipient cells with mCherry.
(MP4)

## Acknowledgments

We thank R. Klein, U. Herian and L. Verhoye for excellent technical assistance and M.T. Pham and M. Cortese for help analyzing the EM data. We are grateful for R. De Francesco for providing the CKIα inhibitor H479 and for GlaxoSmithKline for providing the PI4KA inhibitor G1. We thank T. von Hahn and S. Ciesek for the pWPI-BLR SEC14L2 (isoform 1) construct, T. Wakita for the JFH-1 isolate, J. Bukh for the DBN3acc isolate, D. Trono for retroviral expression constructs, I. Ambiel and O. Fackler for SG-PERT RT standards and C. Rice for Huh7.5 cells and the 9E10 antibody. We also thank the Infectious Disease Imaging Platform (IDIP) headed by V. Laketa for facility use and help with microscopy. We are grateful to the Electron Microscopy Core Facility (Heidelberg University) headed by S. Hillmer for providing access to their equipment and for excellent support.

## Author Contributions

**Conceptualization:** Volker Lohmann.

**Investigation:** Christian Heuss, Paul Rothhaar, Rani Burm, Ji-Young Lee, Philipp Ralfs, Uta Haselmann, Ombretta Colasanti, Cong Si Tran, Noemi Schäfer.

**Methodology:** Christian Heuss, Paul Rothhaar, Frederik Graw, Vibor Laketa.

**Resources:** Paul Schnitzler, Uta Merle, Ralf Bartenschlager, Arvind H. Patel, Thomas Krey, Philip Meuleman.

**Software:** Frederik Graw, Vibor Laketa.

**Supervision:** Philip Meuleman, Volker Lohmann.

**Visualization:** Luisa J. Ströh.

**Writing – original draft:** Christian Heuss, Paul Rothhaar, Volker Lohmann.

**Writing – review & editing:** Christian Heuss, Paul Rothhaar, Rani Burm, Ji-Young Lee, Philipp Ralfs, Uta Haselmann, Luisa J. Ströh, Ombretta Colasanti, Cong Si Tran, Noemi Schäfer, Paul Schnitzler, Uta Merle, Ralf Bartenschlager, Arvind H. Patel, Frederik Graw, Thomas Krey, Vibor Laketa, Philip Meuleman, Volker Lohmann.

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
