## [Decision Letter · Decision Letter 0]

9 May 2022

Dear Volker,

Thank you very much for submitting your manuscript "A Hepatitis C virus genotype 1b post transplant isolate with high replication efficiency in cell culture and its adaptation to infectious virus production in vitro and in vivo" for consideration at PLOS Pathogens. As with all papers reviewed by the journal, your manuscript was reviewed by members of the editorial board and by several independent reviewers. The reviewers appreciated the attention to an important topic. the quality of the work, and the impact of the study for the HCV field. Based on the reviews, we are likely to accept this manuscript for publication, providing that you modify the manuscript according to the reviewer's recommendations.

Sincerely,

Charles M Rice

Associate Editor

PLOS Pathogens

Guangxiang Luo

Section Editor

PLOS Pathogens

Kasturi Haldar

Editor-in-Chief

PLOS Pathogens

orcid.org/0000-0001-5065-158X

Michael Malim

Editor-in-Chief

PLOS Pathogens

orcid.org/0000-0002-7699-2064

Your manuscript has been reviewed by two experts in the field. Both were highly positive of the quality of the study and its impact. However, the reviewers did raise several points that they felt were important for finalizing the submission.

Reviewer Comments (if any, and for reference):

Reviewer's Responses to Questions

**Part I - Summary**

Reviewer #1: See the attached file.

Reviewer #2: The study by Heuss et al. (PPATHOGENS-D-22-0053), describes a novel cloned hepatitis C genotype 1b strain with intrinsic high levels of replication in cell culture. The authors provide evidence for the mechanism behind this unique phenotype. Through a remarkable process of culture adaptation, the authors then succeed in selecting mutations that overcome a restriction on infectious particle production and perform exhaustive mechanistic analysis. Finally, experiments in human liver chimeric mice show that the cell culture adapted clone retains viability in vivo, and importantly identify a critical mutation in NS5A. Thus, this is the first paper to demonstrate in vivo viability of a highly adapted HCVcc; most existing full-length culture systems depend on a large number of adaptive mutations.

This work is original, highly relevant and represents a significant advance in the field, since most HCV isolates are non-replicative in vitro and genotype 1b, the most prevalent genotype worldwide, have been notoriously difficult to culture. Further, it has important in vivo data. However, there are a few issues that needs to be addressed.

Specific comments.

1. The authors should acknowledge/cite the previous work by Jinqian Li et al., who developed genotype 1b cell culture systems (https://doi.org/10.1016/j.antiviral.2021.105136), and therefore tone down or delete statements referencing GLT1cc as the first efficient infectious cell culture system for genotype 1b throughout the text, including in the abstract.

2. The following statement from lines 75-77: “It also provides novel perspectives towards our understanding how liver transplantation drives the evolution of viral isolates with high replication capacity, which might contribute to direct pathogenesis of HCV infection.”

Although certainly a very relevant topic, the authors should tone down the contribution of the study to the understanding of evolution of viruses and liver transplantation, as none of the experiments and data presented address this aspect.

3. Were the experimental conditions applied in this study identical to those from references #17 and #30? What is the variability of methods assessing the size of DMVs? In other words, can the authors reliably compare the size of GLT1 DMVs with data on Con1 and JFH1 obtained from previous studies? If not, DMVs of transfected cells with Con1, JFH1 and GLT1 replicons should be measured head-to-head.

4. Lines 313-315, the origin of the GLT1-mut4B sample is missing.

5. In Fig 4C, can the residual level of Jc1 spread in CD81 deficient cells be explained?

6. Lines 323-325: “comparing GLT1-20M with JC1 324 and the cell culture adapted gt3a isolate DBN3acc (36), generating similar titers as GLT1-20M (Fig. S6A).”

Could the authors clarify in which conditions were the titers of GLT1-20M and DBN3acc comparable? In Fig.S6A the titers of GLT1-20M present high variability in case that the depicted bar is indeed an error bar, which should be stated in the figure legend.

7. In the legend of figure 4D, please specify time point measured.

8. In figure 4E, correct the typo, both cell lines have the same name.

9. Abstract: “but cell-to-cell spreading efficiency was not higher as in other isolates like JFH-1”

Suboptimal language – rewrite.

10. Introduction: “Meanwhile, additional genomes capable of infectious virus production based on gt1a, gt2, gt3a and gt6a have been generated (reviewed in (26)).”

Update to include gt4a, which were recently published also. Subtypes not specified for gt2 (2a, 2b and 2c have been cultured)

11. Introduction: “GLT1cc therefore closes an important gap in the availability of full-replication cycle models of all major HCV genotypes”

Since there is already published full-length culture systems for at least 3 gt1a and 2 gt1b isolates, this statement should be changed to reflect this fact.

12. Results: “We therefore PCR-amplified the viral genomic sequences from infected Huh7 cells and generated a consensus sequence based on direct sequencing of the PCR products”

The authors should specify here that the 5’ and 3’ terminal sequences were derived from Con1 (and not determined from the patient sequence).

13. Discussion: “So far they have been established for gt1a, gt2a, gt2b, gt3a and gt6a, with varying efficiency, requiring up to 20 mutations for efficient virus production (reviewed in (26)), but no efficient gt1b isolate is available.”

Update to reflect comments above about HCVcc for genotype 4a and other genotype 1b isolates.

**Part II – Major Issues: Key Experiments Required for Acceptance**

Reviewer #1: See the attached file.

Reviewer #2: (No Response)

**Part III – Minor Issues: Editorial and Data Presentation Modifications**

Reviewer #1: See the attached file.

Reviewer #2: (No Response)

PLOS authors have the option to publish the peer review history of their article (what does this mean?). If published, this will include your full peer review and any attached files.

Reviewer #1: No

Reviewer #2: No

Figure Files:

Data Requirements:

Reproducibility:

References:

---

## [Editor Report · Decision Letter 1]

29 May 2022

Dear Volker,

Many thanks for your thoughtful responses to the reviewers' critiques and the revised manuscript incorporating these changes. We are pleased to inform you that your manuscript 'A Hepatitis C virus genotype 1b post transplant isolate with high replication efficiency in cell culture and its adaptation to infectious virus production in vitro and in vivo' has been provisionally accepted for publication in PLOS Pathogens.

Best regards,

Charles M Rice

Associate Editor

PLOS Pathogens

Guangxiang Luo

Section Editor

PLOS Pathogens

Kasturi Haldar

Editor-in-Chief

PLOS Pathogens

orcid.org/0000-0001-5065-158X

Michael Malim

Editor-in-Chief

PLOS Pathogens

orcid.org/0000-0002-7699-2064

---

## [Editor Report · Acceptance letter]

20 Jun 2022

Dear Dr. Lohmann,

We are delighted to inform you that your manuscript, "A Hepatitis C virus genotype 1b post-transplant isolate with high replication efficiency in cell culture and its adaptation to infectious virus production in vitro and in vivo," has been formally accepted for publication in PLOS Pathogens.

Best regards,

Kasturi Haldar

Editor-in-Chief

PLOS Pathogens

orcid.org/0000-0001-5065-158X

Michael Malim

Editor-in-Chief

PLOS Pathogens

orcid.org/0000-0002-7699-2064